# On-liquid surface synthesis of diyne-linked two-dimensional polymer crystals

Ye Yang[1], Yufeng Wu[2], Chang Liu [3,4], Mike Hambsch [5], Tiange Dong [2], David Bodesheim [6], Mahabir Prasad[7,8], Arezoo Dianat [6], Thomas D. Kühne[7,8,9], Gianaurelio Cuniberti [6,10], Stefan C. B. Mannsfeld [5], Stuart S. P. Parkin [2], Renhao Dong [11,12], Zhiyong Wang[1,2] ✉ & Xinliang Feng[1,2] ✉

The synthesis of thin crystalline two-dimensional polymers (2DPs) typically relies on reversible dynamic covalent reactions. While substantial progress has been made in solution-based and interfacial syntheses, achieving 2DPs through irreversible carbon-carbon coupling reactions remains a formidable challenge. Herein, we present an on-liquid surface (a mixture of N,N-dimethylacetamide and water, DMAc-H$_2$O) synthesis method for constructing diyne-linked 2DP (DY2DP) crystals via Glaser coupling, assisted by a perfluoro-surfactant (PFS) monolayer. In-situ spectroscopic and diffraction techniques reveal that the well-ordered PFS monolayer facilitates the accumulation of Cu$^+$ ions and subsequent vertical coupling of acetylenic monomers on the DMAc-H$_2$O surface. Building on these findings, we successfully synthesized micro-scale rod-shaped DY2DP-Por or graphdiyne (GDY) crystals through the polymerization of porphyrin- or benzene-based monomers, respectively. Our study represents a significant advancement in the field of on-liquid surface chemistry and opens up enormous opportunities for constructing C−C bond linked 2DP crystals with unique functionalities.

Two-dimensional polymers (2DPs) and their layer-stacked 2D covalent organic frameworks (2D COFs) are an emerging class of covalently bonded polymeric networks with well-defined, periodic structures extending in two distinct directions[1–4]. These layered crystalline materials are predominantly synthesized under solvothermal conditions[5] via dynamic covalent chemistry (DCC)[6], such as reversible Schiff base[7–9], boronic acid condensations[10,11], and quasi-reversible Knoevenagel reactions[12,13], which exploit the error-correction mechanism to promote the formation of periodic polymer network structures[14]. In contrast, the synthesis of 2DPs via irreversible reactions —particularly carbon-carbon (C−C) coupling reactions[15,16], which endow the resulting 2DPs with excellent chemical stability, unique electronic structures and physical properties[17]—remains in its infancy. Due to the lack of error-correction capability, achieving high crystallinity in these 2DPs using solution-based synthesis methods is extremely challenging.

[1]Center for Advancing Electronics Dresden & Faculty of Chemistry and Food Chemistry, Technische Universität Dresden, Dresden, Germany. [2]Max Planck Institute of Microstructure Physics, Halle (Saale), Germany. [3]MOE Engineering Research Center of Membrane and Water Treatment, and Key Lab of Adsorption and Separation Materials & Technologies of Zhejiang Province, Department of Polymer Science and Engineering, Zhejiang University, Hangzhou, China. [4]The "Belt and Road" Sino-Portugal Joint Lab on Advanced Materials, International Research Center for X Polymers, Zhejiang University, Hangzhou, China. [5]Center for Advancing Electronics Dresden & Faculty of Electrical and Computer Engineering, Technische Universität Dresden, Dresden, Germany. [6]Institute for Materials Science and Max Bergmann Center for Biomaterials, TUD Dresden University of Technology, Dresden, Germany. [7]Center for Advanced Systems Understanding, Görlitz, Germany. [8]Helmholtz-Zentrum Dresden-Rossendorf, Dresden, Germany. [9]Institute of Artificial Intelligence, Chair of Computational System Sciences, Technische Universität Dresden, Dresden, Germany. [10]Dresden Center for Computational Materials Science (DCMS), TUD Dresden University of Technology, Dresden, Germany. [11]Department of Chemistry, The University of Hong Kong, Hong Kong, China. [12]Materials Innovation Institute for Life Sciences and Energy (MILES), HKU-SIRI, Shenzhen, China. ✉e-mail: zhiyong.wang@mail.mpi-halle.mpg.de; xinliang.feng@mpi-halle.mpg.de

To overcome the above limitations, a range of interfacial synthetic methodologies, such as on-metal surface synthesis[18–20], on-water surface synthesis[21–24], and liquid-liquid interfacial synthesis[25], have been developed. In contrast to solution-based synthesis, these approaches offer a confined 2D space that directs the programmable pre-organization of monomers and enhances the reactivity[26,27], facilitating the growth of well-ordered frameworks[28]. Among them, the on-metal surface method has been particularly successful in synthesizing crystalline 2D conjugated polymers through irreversible C−C coupling reactions. Notable examples include the on-metal surface synthesis of 2D polyphenylene via Ullmann coupling[29,30], graphyne via Sonogashira coupling[31,32] and graphdiyne (GDY) via Glaser coupling[33,34]. Nevertheless, the strong interaction between monomers and the reactive metal surface limits the long-range growth of 2DP crystal domains and complicates their subsequent transfer to other substrates for characterization and device integration[35]. While the on-water surface and liquid-liquid interface would benefit from weak monomer-surface interactions[36], the low reactivity of C−C coupling reactions in these systems hampers the efficiency of 2D polymerization[37]. Despite significant advances, synthesizing crystalline 2DPs via irreversible C−C coupling reactions remains a formidable challenge due to the intrinsic incompatibility between monomer assembly and reaction reactivity.

In this study, we demonstrate an on-organic liquid surface synthesis strategy driven by the perfluoro-surfactant (PFS) monolayer on the surface of a N,N-dimethylacetamide/water (DMAc-H$_2$O) mixture (named as O-SMAIS), which allows for constructing diyne-linked 2DP (DY2DP) crystals via Glaser coupling reactions. In-situ polarization modulation infrared reflection-absorption spectroscopy (PM-IRRAS)[38,39] and X-ray diffraction and fluorescence[40,41] measurements revealed that the PFS monolayer (a = b = 5.8 Å, γ = 120°) formed on the DMAc-H$_2$O surface promotes the accumulation of Cu$^+$ ions via electrostatic interactions, creating a catalyst-rich surface (Cu$^+$ concentration, 265 mM). Subsequently, the strong coordination between terminal alkynes and Cu$^+$ ions drives the assembly of acetylenic monomers on the DMAc-H$_2$O surface. Model reactions and theoretical calculations demonstrated that the alkynes can be highly activated by the accumulated Cu$^+$ ions, leading to the formation of diyne bonds with enhanced reactivity compared to the bulk solution. Building on this finding, we successfully synthesized rod-shaped DY2DP (DY2DP-Por) and GDY crystals with micron sizes from 5,10,15,20-tetrakis(4-ethynylphenyl)porphyrin (TEPP) and 1,2,3,4,5,6-hexakis((-trimethylsilyl)ethynyl)-benzene (HEB-TMS), respectively. X-ray diffraction (XRD) and transmission electron microscopy (TEM) resolved their crystal structures, revealing lattice parameters of a = b = 22.9 Å, c = 7.3 Å, α = γ = 90°, β = 60° for DY2DP-Por, and a = b = 9.5 Å, c = 3.4 Å, α = 73.5°, β = 90°, γ = 120° for GDY. Notably, the resulting DY2DP-Por and GDY crystals can be mechanically exfoliated into 2D nanoflakes with lateral sizes of ~0.02 and 0.6 μm$^2$, and thicknesses of ~4.5 and 8.4 nm, respectively. We further demonstrated the unique interlayer semiconducting nature of the newly developed DY2DP-Por through electrical measurements on crystal devices. These findings push the boundaries of on-liquid surface chemistry and unlock vast potential for developing 2DP crystals via irreversible coupling reactions.

## Results

### On-liquid surface chemistry

To conduct reactions on a liquid surface, the liquid must have sufficient surface tension (e.g., water, 72.8 mN m$^{-1}$)[42] to support the molecules (Supplementary Figs. 1 and 2) while ensuring their solubility[43]. Most organic solvents, such as chloroform, acetone, and ethanol, have a surface tension below 35 mN m$^{-1}$, which is insufficient for stable support[44], even for DMAc (35.2 mN m$^{-1}$)[45]. A combination of interface properties (e.g., surface tension, interfacial hydrogen bonding, etc.) collectively governs the selection of a mixture of DMAc and H$_2$O (v:v = 1:1, named as DMAc-H$_2$O) as solvent, which offers both a

higher surface tension of 57.7 mN m$^{-1}$ (Supplementary Fig. 3) and good solubility for alkynyl monomers. The observed variations in surface tension at different mixing ratios of the bulk phase of DMAc-H$_2$O suggest that the surface composition of the mixture closely aligns with the bulk phase. Instead of surfactants with an alkyl chain (e.g., stearic acid, SA), which are unstable on the DMAc-H$_2$O surface, we employed PFS with fluorinated alkyl chains, and polar head groups (i.e., -COOH) to construct the surfactant monolayer (Fig. 1a, b). The in-situ grazing incidence wide-angle X-ray scattering (GIWAXS) pattern exhibits five sharp and discrete diffraction peaks at $Q_{xy}$ = 1.26, 2.20, 2.54, 3.36, and 3.82 Å$^{-1}$, corresponding to the (100), (110), (200), (210), and (300) planes of assembled PFS monolayer, respectively, in a polytetrafluoroethylene (PTFE) trough (Fig. 1c). This result elucidates the long-range ordered structure of PFS monolayer on the DMAc-H$_2$O surface with a unit cell of a = b = 5.8 Å, γ = 120° (Fig. 1c, inset Supplementary Fig. 4). The PFS monolayer remains stable on the DMAc-H$_2$O surface even after 2 days as evidenced by the constant surface pressure in Supplementary Figs. 4−6, which is sufficient to guide the subsequent assembly and polymerization of monomers[46].

The three-step procedure for the O-SMAIS strategy is illustrated in Fig. 1d. First, a PFS (3.1×10$^{-8}$ mol) monolayer was prepared on the DMAc-H$_2$O surface in a crystallization dish (diameter, 6 cm, Step I). Next, a CuCl aqueous solution (3×10$^{-5}$ mol), as the catalyst of Glaser coupling, was added to the liquid phase. Due to strong electrostatic interactions, the Cu$^+$ ions were accumulated underneath the anionic PFS monolayer to form the catalyst-rich surface (Step II). Then, acetylenic monomers dissolved in DMAc were injected into the subphase. The strong coordination between terminal alkynes and Cu$^+$ promoted the vertical assembly of acetylenic monomers, thereby initiating the Glaser coupling reaction on the DMAc-H$_2$O surface (Step III). After 24 h, we employ a dipping process where the substrate (e.g., SiO$_2$/Si wafer, TEM grid) is carefully positioned beneath the 2DPs floating on the DMAc-H$_2$O surface and then lifted horizontally (Supplementary Fig. 7).

### Reactivity of glaser coupling on the DMAc-H$_2$O surface

To evaluate the reactivity of Glaser coupling on the DMAc-H$_2$O surface, we performed a model reaction (Model I) using 10-(4-ethynylphenyl)−5,15,20-triphenylporphyrin (**1**) using the O-SMAIS strategy (Fig. 2a, and Supplementary Figs. 8 and 9). After 24 h, the products from both the surface and bulk solution were collected and analyzed by matrix-assisted laser desorption/ionization-time-of-flight mass spectrometry (MALDI-TOF MS). As shown in Fig. 2b, the MS spectrum of the compound in solution presents an apparent peak at m/z = 699.161, corresponding to the unreacted **1a**. This suggests that the model reaction scarcely proceeds in the solution phase (Supplementary Figs. 10 and 11). In contrast, for the product on the DMAc-H$_2$O surface, the MS peak of monomer **1** vanishes, and a new peak belonging to the final product **2** appears at m/z = 1396.308, indicating significantly higher reactivity of Glaser coupling on the DMAc-H$_2$O surface than in the bulk solution. To validate this finding, Model II was carried out employing phenylacetylene (**3**) under identical conditions (Fig. 2c). The target product was exclusively obtained on the DMAc-H$_2$O surface, as identified by nuclear magnetic resonance (NMR) spectroscopy, further supporting the enhanced reactivity compared to the bulk solution (Supplementary Fig. 12).

To elucidate the crucial role of the DMAc-H$_2$O surface in enhancing the reactivity of Glaser coupling reactions, we calculated the reaction energy barriers on the DMAc-H$_2$O surface and in bulk solution using the density functional theory (DFT) method (Fig. 2d, Supplementary Figs. 13 and 14). The Glaser coupling involves three steps[47]: coordination (**3**–**3a**), oxidation (**3a**–**3b**), and reductive elimination (**3b**–**4**). Notably, the reactions on the DMAc-H$_2$O surface exhibit lower energy barriers for the Cu$^+$-acetylide **3a** (−4.27 vs. −3.51 eV), intermediate **3b** (−2.65 vs. −2.18 eV) and final product **4** (−6.10 vs. −0.13 eV) than those in bulk solution (Fig. 2e). The reduced energy barrier at **3a**

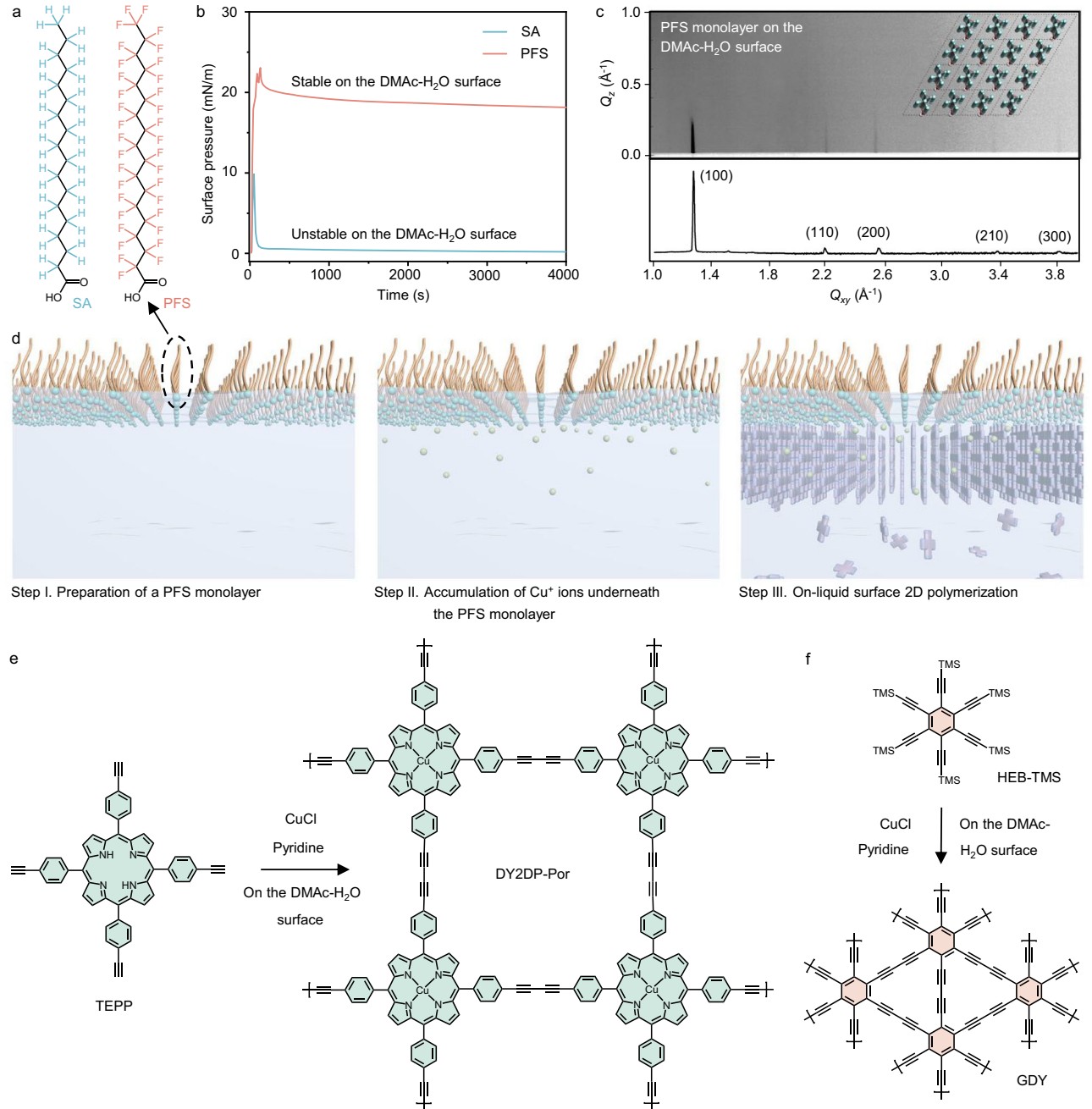

Fig. 1 | Synthesis protocol of DY2DPs. a Chemical structures of stearic acid (SA) and perfluoro-surfactant (PFS). b Surface pressures of the DMAc-H$_2$O surface during the formation of SA or PFS monolayer. c In-situ GIWAXS pattern and its in-plane projection (near $Q_z = 0$, $Q$ represents scattering vector) of PFS monolayer in Step I. Inset: Simulated structure of the resulting PFS monolayer. d Schematic illustration of the synthetic procedure for DY2DPs on the DMAc-H$_2$O surface. e, f Schematic of the 2D polymerization of TEPP to construct DY2DP-Por with a square lattice (e), as well as the polymerization of HEB-TMS to construct GDY with a hexagonal lattice (f).

($\Delta G_1 = -0.76$ eV) demonstrates the enhanced ability of Cu$^+$ catalyst accumulated on the DMAc-H$_2$O surface to activate alkynes. Besides, the significant decrease in the energy barrier of 4 ($\Delta G_2 = -5.97$ eV) suggests that the 2D confinement at the interface minimizes out-of-plane motion and steric hindrance of compound 3b, facilitating the formation of diyne bonds on the DMAc-H$_2$O surface.

### Real-time study of 2D polymerization and crystallization on the DMAc-H$_2$O surface

Building on the experimental and theoretical findings from model reactions, we proceeded to carry out the synthesis of DY2DP-Por

through the 2D polymerization of TEPP on the DMAc-H$_2$O surface (Fig. 1e). To gain deeper insight into this interfacial strategy, we performed in-situ PM-IRRAS, X-ray fluorescence (XRF), GIWAXS, and grazing incidence small-angle X-ray scattering (GISAXS) measurements from Step I to Step III (Fig. 3a and Supplementary Figs. 15 and 16). In Step I, a long-range ordered PFS monolayer has been demonstrated by in-situ GIWAXS (Fig. 1c). In-situ PM-IRRAS spectra show that a broad absorption peak at ~3680 cm$^{-1}$, corresponding to the -OH stretching, was observed in the pure DMAc-H$_2$O system (Supplementary Fig. 17). Upon addition of the PFS monolayer, this peak undergoes significant decreases in intensity, due to the surface

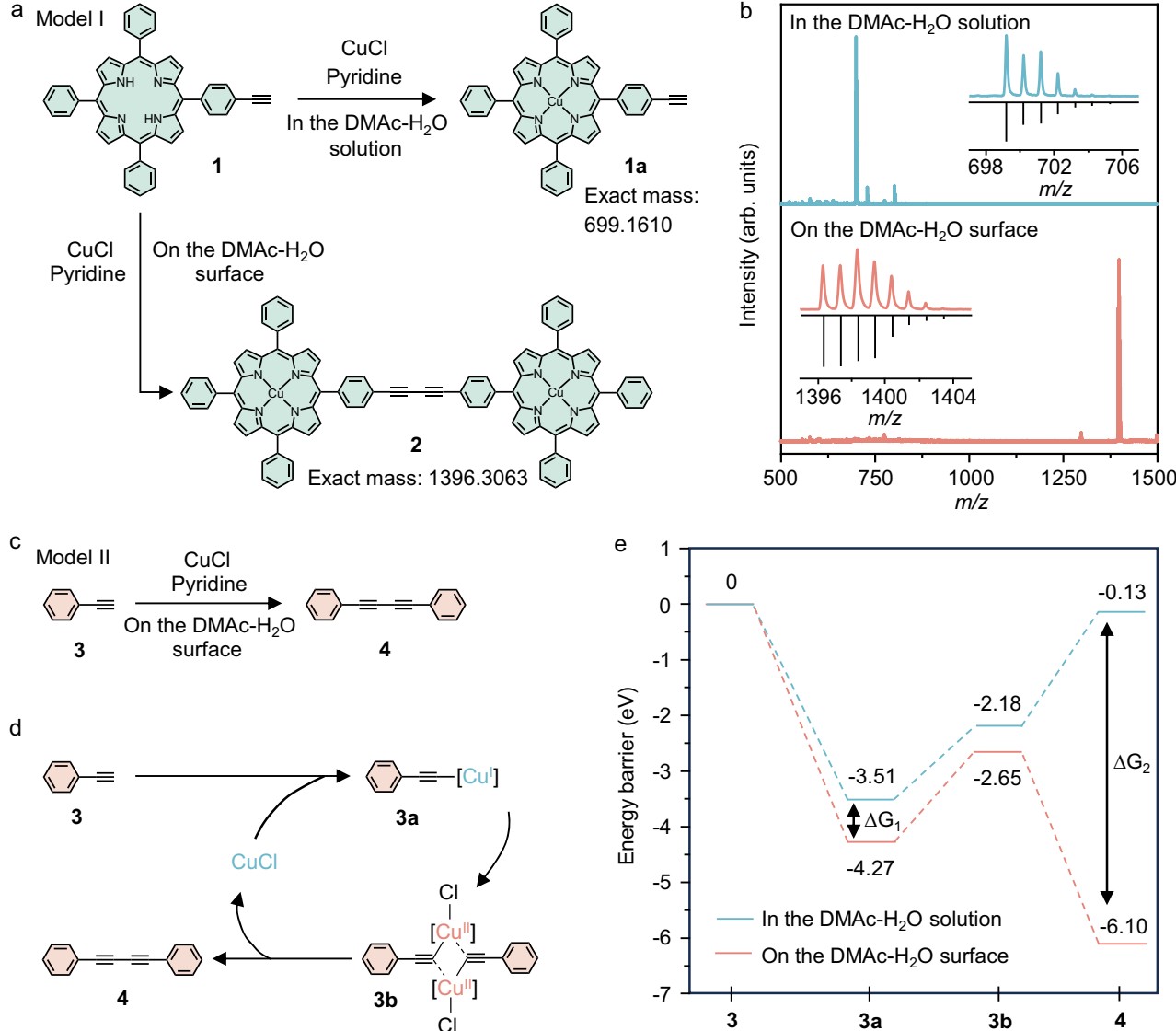

**Fig. 2 | Model reactions on the DMAc-H$_2$O surface. a**, **c** Schematic of Model I (**a**) and Model II (**c**) reactions. Compounds **1**, **3** yield **2**, **4**, respectively. **b** MALDI-TOF MS analysis of the Model I products on the DMAc-H$_2$O surface (bottom) and in bulk solution (top). Insets: High-resolution MALDI-TOF MS spectra of these products.

**d**, **e** Mechanism (**d**) and its calculated energy profiles (**e**) of the Glaser coupling reaction. The energy barriers are with respect to **3**. The energy barrier differences for the reactions in the DMAc-H$_2$O solution and on the surface in **3a** and **4** are denoted as $\Delta G_1$ and $\Delta G_2$.

crowding[48] and water structure fluctuation[49]. The integral intensity of C-F vibration peaks at ~1207 and 1151 cm$^{-1}$ remains consistent after 4 h (Fig. 3b, 1$^{st}$-3$^{rd}$ curves), indicating the high ordering and stability of the PFS monolayer on the DMAc-H$_2$O surface.

In Step II, the negatively charged head group of PFS promotes the electrostatically driven accumulation of positively charged Cu$^+$ (forming PFS-Cu). The concentration of Cu$^+$ accumulated beneath the PFS monolayer was determined by in-situ XRF. As shown in Fig. 3c,f, and Supplementary Fig. 18, after adding CuCl solution for 4 h, the concentration of Cu$^+$ at the interface was observed to increase from 0 to 265 mM, which is ~160 times higher than that of the reference experiment (i.e., CuCl solution without PFS on the surface, 1.5 mM, Supplementary Figs. 19–21). Moreover, the in-situ GIWAXS and PM-IRRAS demonstrate that the PFS monolayer retains its crystallinity and stability during the accumulation process (Fig. 3b, 4$^{th}$-5$^{th}$ curves, Supplementary Figs. 22 and 23). In this step, the Cu$^+$-rich DMAc-H$_2$O surface was formed, allowing for simultaneously guiding the subsequent vertical assembly and polymerization of acetylenic monomers.

In Step III, after adding TEPP monomers for 2 h, a new PM-IRRAS peak was observed at 3264 cm$^{-1}$, corresponding to the C≡C-H stretching in the monomer (Fig. 3b, 6$^{th}$ curve). This result reveals the rapid accumulation of TEPP from the bulk DMAc-H$_2$O solution onto its surface due to the strong coordination between the terminal alkynes and the accumulated Cu$^+$. Besides, two weak signals appearing at $Q_z$ = 0.43 and 0.54 Å$^{-1}$ in the GISAXS spectrum correspond to the DY2DP-Por planes with $d$-spacings of 14.6 Å and 11.6 Å, which confirms its vertically oriented nucleation process (a crystal nucleus size of ~20 nm) on the DMAc-H$_2$O surface (Fig. 3d, e, 6$^{th}$ curve). The DFT calculation shows that the vertically oriented alignment of TEPP underneath a PFS monolayer possesses lower energy than its face-on structure (Supplementary Fig. 24), agreeing well with the experimental results. The reaction was kept for 14 h to record the growth of DY2DP-Por crystals in real-time. PM-IRRAS spectra demonstrate an obvious decrease in the surface content of C≡C-H in TEPP (Fig. 3b, 6$^{th}$-12$^{th}$ curves, 3 g). After the coupling of the terminal alkynes, XRF spectra show that the accumulated Cu$^+$ migrates downward from the surface layer (Fig. 3f), continuously

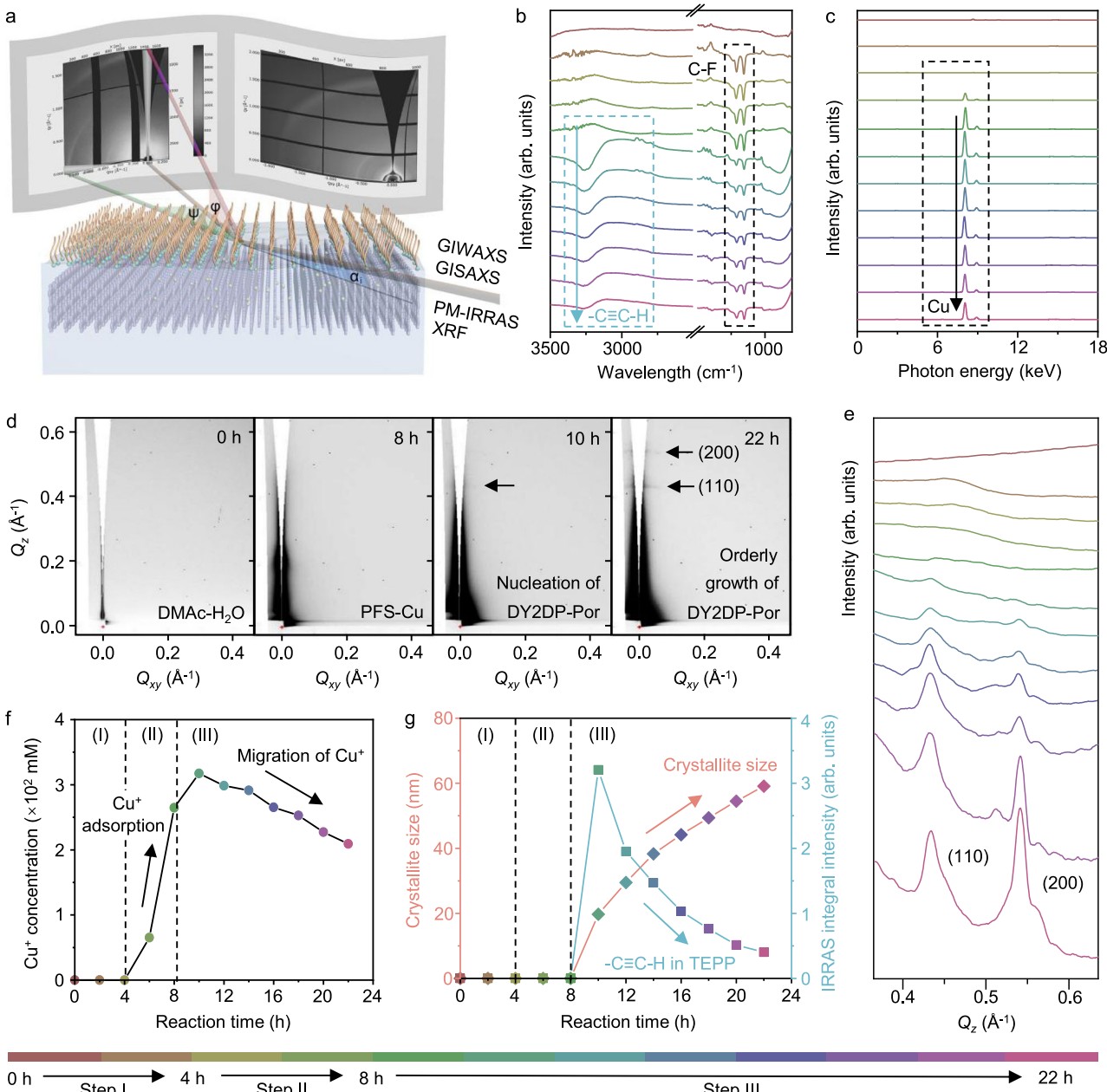

**Fig. 3 | In-situ studies of the interfacial growth of DY2DP-Por. a** Schematic illustration of in-situ step-by-step measurements for the 2D polymerization on the DMAc-H₂O surface, including grazing incidence wide-angle X-ray scattering (GIWAXS), grazing incidence small-angle X-ray scattering (GISAXS), polarization modulation infrared reflection-absorption spectroscopy (PM-IRRAS) and X-ray fluorescence (XRF). **b** Time evolution of the PM-IRRAS spectra with an angle of incidence $\alpha_i = 60°$ in p-polarization recorded on the DMAc-H₂O surface from Step I to Step III. **c** Time evolution of the XRF spectra recorded on the DMAc-H₂O surface from Step I to Step III. **d** Representative in-situ GISAXS patterns for Step I-III measured on the DMAc-H₂O surface. **e** Time evolution of the out-of-plane GISAXS projections (near $Q_{xy} = 0$, $Q$ represents scattering vector) during Step I-III. **f** Calculated concentration of Cu⁺ on the DMAc-H₂O surface through the peak at the photon energy of 8.04 eV in the in-situ XRF spectra. **g** Time-dependent PM-IRRAS integral areas of the vibrational peak of -C≡C-H (3264 cm⁻¹) for p-polarization as well as the calculated crystallite size of DY2DP-Por using the full width at half maximum of the Gaussian fit of the peak at $Q_z = 0.54$ Å⁻¹ in the in-situ GISAXS spectra. Color scale from red to purple indicates on-water surface synthesis durations ranging from 0 to 22 h.

catalyzing the longitudinal polymerization. Note that the GISAXS peak intensities of the DY2DP-Por planes gradually increase in the $Q_z$ direction (Fig. 3d, e, 6th–12th curves, and Supplementary Fig. 25). The crystallite size of DY2DP-Por, as determined by the Scherrer equation[50]: $D = K\lambda/(\beta\cos\theta)$ (where K is a constant of 0.94; $\lambda$ is the beam wavelength; $\beta$ is the full width at half maximum; $\theta$ is the diffraction angle), increases from ~0 to 60 nm after 14 h (Fig. 3g). This result indicates that the in-plane structure of the oriented crystal

nucleus expands in an orderly manner perpendicular to the DMAc-H₂O surface. A comparative analysis of these two lattice structures reveals partial crystallographic alignment, such as the (100) plane of the PFS monolayer (d-spacing of ~0.50 nm) show lattice matching with the (040) plane of DY2DP-Por (d-spacing of ~0.50 nm) (Supplementary Fig. 26). This observation supports the notion that DY2DP-Por crystals grow epitaxially beneath the long-range ordered PFS monolayer (Supplementary Figs. 27–31)[51].

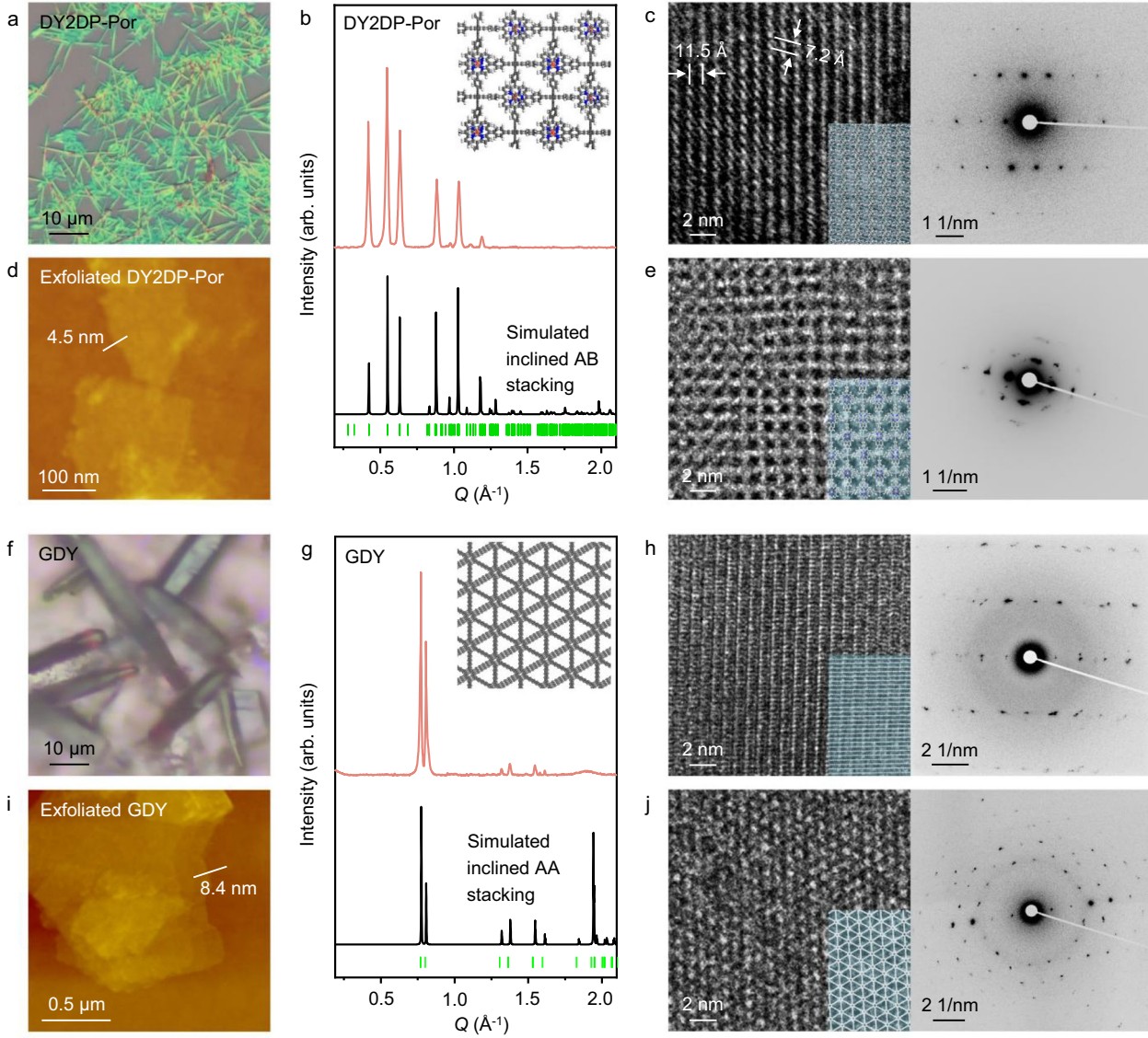

**Fig. 4 | Structural characterizations of DY2DP-Por and GDY. a** OM image of DY2DP-Por crystals. **b** XRD patterns of DY2DP-Por crystals and the simulation based on inclined AB stacking. Inset: the simulated AB-stacked structure from the top view. **c** HRTEM images and the SAED pattern of rod-shape DY2DP-Por crystals overlaid with structure modes (bottom inset). The (200) plane direction is not perpendicular to the (001) plane, indicating its inclined stacking mode. **d**, **e** AFM (**d**), HRTEM images and SAED pattern (**e**) of exfoliated DY2DP-Por flakes overlaid with structure modes (bottom inset). Thickness along the white line: 4.5 nm. **f** OM image of GDY crystals. **g** XRD patterns of GDY crystals and the simulation based on inclined AA stacking. Inset: the simulated AA-stacked structure from the top view. **h** HRTEM images and the SAED pattern of rod-shape GDY crystals overlaid with structure modes (bottom inset). **i**, **j** AFM (**i**), HRTEM images and SAED pattern (**j**) of exfoliated GDY flakes overlaid with structure modes (bottom inset). Thickness along the white line: 8.4 nm.

## Structural characterization of DY2DP-Por crystals

After 24 h of polymerization, the DY2DP-Por crystals formed on the DMAc-H$_2$O surface were horizontally transferred onto various substrates (e.g., SiO$_2$/Si wafers, gold foils, quartz plates, TEM grids, etc.) for morphological and structural characterizations. Polarizing optical microscopy (OM), scanning electron microscopy (SEM), and atomic force microscopy (AFM) reveal rod-shape DY2DP-Por crystals with a length of ~10 μm and a thickness of ~0.5 μm (Fig. 4a and Supplementary Figs. 32 and 43a−c). UV-vis spectrum shows red-shift in the Q band of the TEPP monomer from 422 nm to 434 nm (Supplementary Fig. 33a), suggesting extended conjugation in DY2DP-Por. The C≡C-H peak of TEPP at 3269 cm$^{-1}$ disappears in attenuated total reflection-Fourier transform infrared (ATR-FTIR) spectra (Supplementary Fig. 33b), while the Raman spectra of the crystals present a new peak at 2208 cm$^{-1}$ (corresponding to -C≡C−C≡C-) (Supplementary Fig. 34),

verifying the formation of the diyne linkage and the complete conversion of acetylene groups. The X-ray photoelectron spectroscopy (XPS, Supplementary Figs. 35 and 36) analysis shows the curve-fitted peaks of C=N (285.8 eV), C≡C (285.1 eV), C=C (284.2 eV), pyrrolic N-Cu (401.2 eV), and pyridinic N-Cu (398.0 eV), confirming that Cu$^{2+}$ ions are coordinated with pyridine molecules and the structural center of porphyrin after the polymerization.

To resolve the periodic structure of the resulting DY2DP-Por crystals, we performed XRD and high-resolution TEM (HRTEM) measurements, in combination with the structural simulations. The XRD patterns exhibit Bragg peaks at Q = 0.42, 0.54, 0.63, 0.88, 1.03, and 1.19 Å$^{-1}$ (2θ = 5.91°, 7.69°, 8.92°, 12.45°, 14.56°, and 16.79°), corresponding to the (110), (200), (020), (021), (212) and (141) planes, respectively. These reflections indicate a square unit cell with parameters of a = b = 22.9 Å, c = 7.3 Å, α = γ = 90°, β = 60° for DY2DP-Por,

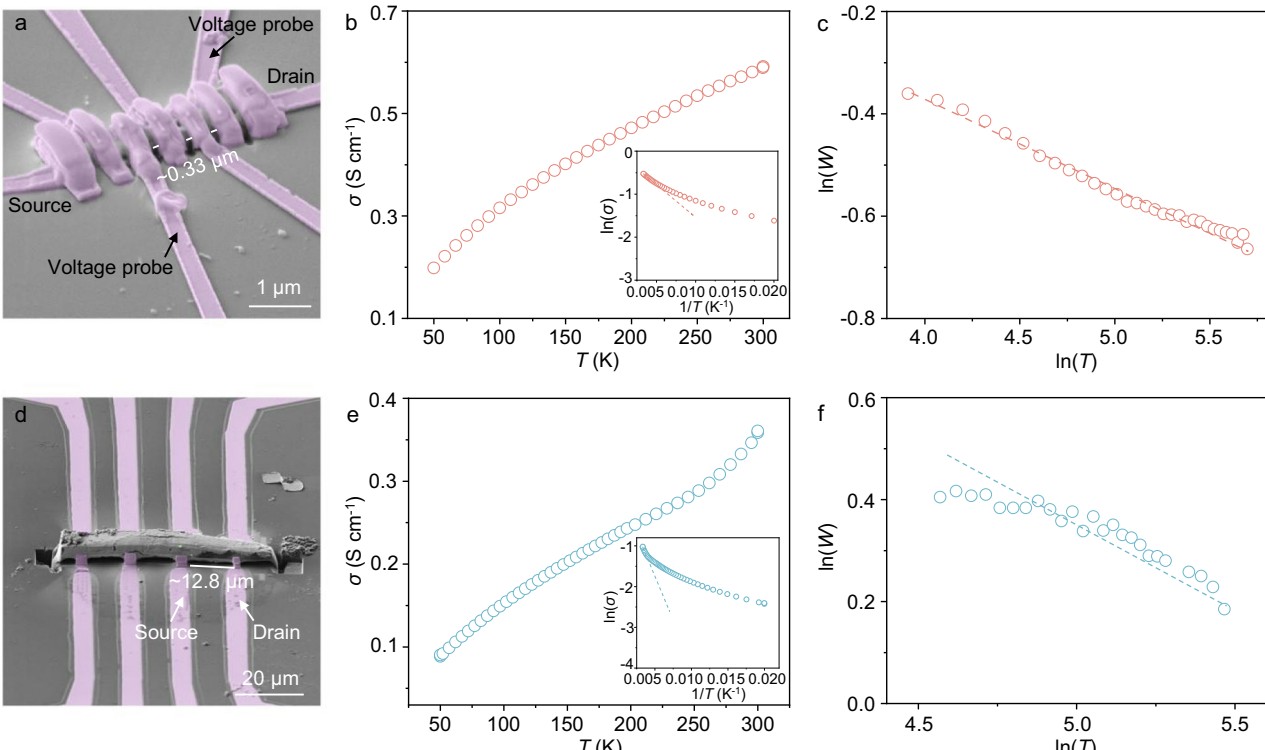

**Fig. 5 | Electronic property of DY2DP-Por and GDY crystals. a, d** SEM images of DY2DP-Por (**a**) and GDY (**d**) crystal device with Pt contacts. **b, e** Temperature-dependent conductivities ($\sigma$) of DY2DP-Por (**b**) and GDY (**e**) crystals ranging from 300 to 50 K. Inset: The plot of $\ln(\sigma)$ versus the reciprocal of the temperature ($1/T$). **c, f** Zabrodskii plots, $\ln(W)$ versus $\ln(T)$, where $W = \mathrm{d}(\ln\sigma)/\mathrm{d}(\ln T)$.

consistent with the in-situ GISAXS results and the calculated crystal structure adopting an inclined AB stacking mode (Fig. 4b and Supplementary Fig. 37). Furthermore, the HRTEM images show the lattice with *d*-spacings of 11.5 and 7.2 Å. The selected area electron diffraction (SAED) patterns also exhibit several sets of diffraction spots (Fig. 4c and Supplementary Fig. 38), where the nearest reflections at 0.88 and 0.14 nm$^{-1}$ are assigned to the (200) and (001) planes of DY2DP-Por, respectively. To characterize the crystal structure within the *ab* plane, we mechanically exfoliated the rod-shape DY2DP-Por crystals, along the *c* direction i.e., π-π stacking direction, into the nanoflakes with a lateral size of ~0.02 μm$^2$ and a thickness of ~4.5 nm (Fig. 4d). As shown in Fig. 4e and Supplementary Fig. 39, the observed tetragonal lattice by HRTEM is in agreement with the calculated structure. These DY2DP-Por crystals exhibit exceptional chemical resistance even in harsh conditions due to their robust diyne linkage (Supplementary Figs. 40 and 41).

### Extending the O-SMAIS strategy towards GDY

To assess the universality of O-SMAIS strategy, we further synthesized GDY using HEB-TMS as the monomer (Fig. 1f), following the Step I, II and III procedures. Under identical conditions, rod-shape GDY crystals with a length of ~30 μm and a thickness of ~5 μm were obtained on the DMAc-H$_2$O surface (Fig. 4f, Supplementary Figs. 42 and 43d–e). The successful 2D polymerization was confirmed by the disappearance of the trimethylsilyl (TMS) peak at ~2961 cm$^{-1}$ in the FTIR spectrum and the appearance of diyne signals at ~1943 and 2158 cm$^{-1}$ in the Raman spectrum (Supplementary Figs. 44–47). The D-band (1384 cm$^{-1}$) and G-band (1561 cm$^{-1}$) intensity ratio of 0.62 indicates high structure periodicity of GDY crystals. The sharp Bragg peaks at $Q$ = 0.77, 0.81, 1.37, 1.54, and 1.88 Å$^{-1}$ ($2\theta$ = 10.86°, 11.35°, 19.42°, 21.84°, and 26.62°) were probed in the XRD spectra, corresponding to the (100), (010), (110), (200) and (001) planes, respectively (Fig. 4g). These results align with the simulated inclined AA-stacked structure with a lattice

parameter of a = b = 9.5 Å, c = 3.4 Å, α = 73.5°, β = 90°, γ = 120° (Supplementary Fig. 48). HRTEM images and SAED patterns present the (100) and (001) planes fringes of rod-shape GDY crystals (Fig. 4h, and Supplementary Fig. 49). To probe the structure of GDY in the *ab* plane, the rod-shape crystals were exfoliated into the nanoflakes with a lateral size of ~0.6 μm$^2$ and a thickness of ~8.4 nm along the π-π stacking direction (Fig. 4i). The highly ordered hexagonal lattice in HRTEM image further supports an inclined AA stacking mode for the obtained GDY (Fig. 4j and Supplementary Fig. 50). A comparative summary of the domain size, orientation and stacking mode of GDY crystals synthesized via the O-SMAIS method and other reported strategies was compiled, including chemical vapor deposition (CVD) methods[52], explosion methods[53], solvothermal synthesis[54], on-metal surface synthesis[33], liquid-liquid interfacial synthesis[55] and gas-liquid interfacial synthesis[36] (Supplementary Table 1). Compared to other strategies, the O-SMAIS method offers precise control over monomer alignment and directional growth of GDY crystals, yielding highly oriented GDY crystals with enlarged domain sizes.

### Electronic property of DY2DP-Por and GDY crystals

DY2DP-Por and GDY crystal devices were fabricated to understand the intrinsic electronic properties along the interlayer direction (Supplementary Figs. 51 and 52). Electron beam lithography (EBL) and focused ion beam (FIB) at ultrahigh vacuum were employed to construct Au/Ti/Pt electrical contacts in a four-point probe measurement geometry (Fig. 5a, d). The linear I-V characteristic across various voltages demonstrates an Ohmic contact between the crystals and gold (Au)/titanium (Ti)/platinum (Pt) wires (Supplementary Figs. 53 and 54). Variable-temperature conductivity measurements of DY2DP-Por and GDY crystals exhibit a nonlinear decline in conductivity from 300 to 50 K (Fig. 5b, e), indicative of the semiconducting nature (Supplementary Figs. 55 and 56). The conductivities ($\sigma$) of DY2DP-Por and GDY crystals were estimated

through Ohm's Law:

$$\sigma = \frac{1}{\rho} = \frac{L}{RA} \qquad (1)$$

where $\rho$ is the resistivity, $R$ is the measured resistance, $L$ is the crystal length between Au/Ti/Pt wires, and $A$ is the cross-sectional area of the crystal. $L$ values were measured to be ~0.33 μm for DY2DP-Por and ~12.8 μm for GDY (Fig. 5a, d). Their $A$ values were roughly estimated to be ~0.3 μm$^2$ for DY2DP-Por and ~40.7 μm$^2$ for GDY. Then, their variable-temperature conductivity curves were obtained as shown in Fig. 5b, e. Notably, the room-temperature $\sigma$ of DY2DP-Por and GDY crystals reaches 0.58 and 0.36 S cm$^{-1}$, respectively, exceeding most reported 2D COFs, 2DPs and GDYs synthesized by other strategies (Supplementary Tables 2 and 3). The high $\sigma$ of DY2DP-Por is likely attributed to pyridine doping during the synthesis. However, for GDY, we like to highlight that correlating conductivity directly to quality or crystallinity remains challenging, as variations in stacking mode (e.g., AA, AB and ABC), layer orientation (e.g., edge-on, face-on and random) and sample forms (e.g., film and crystal) of the materials, also significantly influence the electronic transport properties despite the shared structure. The activation energies ($E_a$) of DY2DP-Por and GDY crystals were estimated using the Arrhenius equation with a logarithmic form:

$$\ln\sigma = \ln\sigma_0 - \frac{E_a}{k_B}\frac{1}{T} \qquad (2)$$

where $\sigma_O$ is the pre-exponential conductivity factor, $k_B$ is the Boltzmann constant, and $T$ is the temperature. $E_a$ of DY2DP-Por and GDY were determined to be ~ 16 meV and ~62 meV, respectively, by plotting the linear form of ln$\sigma$ vs. 1/$T$. This activation barrier reflects the effective energy that charge carriers must overcome when traversing the fluctuating energy landscape in DY2DP-Por, and it is not related to the bandgap. Zabrodskii analysis was employed to further understand the transport regime (Fig. 5c, f). The negative slope in ln($W$) versus ln($T$) plots, where $W = $ d(ln$\sigma$)/d(ln$T$), indicates their hopping-dependent transport.

## Discussion

In summary, we have developed the O-SMAIS strategy for the synthesis of 2D polymer crystals with diyne linkages, including DY2DP-Por and GDY, through the irreversible Glaser coupling reaction on the DMAc-H$_2$O surface. In-situ, IRRAS, X-ray diffraction, and fluorescence highlighted that the well-ordered PFS monolayer on the DMAc-H$_2$O surface promotes the accumulation of Cu$^+$ ions and the formation of a catalyst-rich surface. The coordination between terminal alkynes and Cu$^+$ ions guides the vertically oriented assembly of acetylenic monomers and epitaxial growth of DY2DP crystals on the DMAc-H$_2$O surface. Using this approach, we demonstrated the higher reactivity of Glaser coupling on the surface, and successfully synthesized micrometer-scale rod-shaped DY2DP-Por and GDY crystals with lattice parameters of a = b = 22.9 Å, c = 7.3 Å, α = γ = 90°, β = 60° and a = b = 9.5 Å, c = 3.4 Å, α = 73.5°, β = 90°, γ = 120°. It is important to note that the mechanical exfoliation enables the delamination of these crystals into nanoflakes with thicknesses of ~4.5 and 8.4 nm, respectively. Electrical measurement of the DY2DP-Por crystal demonstrates its unique interlayer semiconducting nature with a room-temperature conductivity of 0.58 S cm$^{-1}$, surpassing most reported 2D COFs and 2DPs. This work represents a significant advance in interfacial chemistry by enabling the synthesis of C−C linked 2DP crystals. In addition, the O-SMAIS strategy would open the door to synthesizing

a wider range of structurally diverse and functional 2DPs with tunable electronic, optical, and mechanical properties.

## Methods

### Synthesis of DY2DP-Por crystals

40 ml of DMAc-Milli-Q water mixture ($v{:}v$ = 1:1) was added into a crystallization dish (diameter, 6 cm). Then, 28 μl of PFS solution (1 mg ml$^{-1}$ in chloroform) was spread on the DMAc-H$_2$O surface using a micropipette to form a surfactant monolayer. After 30 min stabilization, 1 ml of CuCl (0.03 mmol) and pyridine (0.06 mmol) aqueous solution was gently injected into the water subphase using a syringe. After 2 h, metal-free TEPP (4.2 μmol in DMAc solution) was added. The reaction was then kept undisturbed at 1 °C for 24 h. After that, the rod-shaped crystals were obtained and deposited onto the substrate via a horizontal dipping approach. Then, the product was rinsed with flowing Milli-Q water, and ethanol and dried in N$_2$ flow.

### Synthesis of GDY crystals

40 ml of DMAc-Milli-Q water mixture ($v{:}v$ = 1:1) was added into a crystallization dish (diameter, 6 cm). 28 μl of PFS solution (1 mg ml$^{-1}$ in chloroform) was spread on the DMAc-H$_2$O surface using a micropipette to form a surfactant monolayer. After 30 min stabilization, 1 ml of CuCl (0.03 mmol) and pyridine (0.06 mmol) aqueous solution was gently injected into the water subphase using a syringe. After 2 h, HEB-TMS (29 μmol in DMAc solution) was added. The reaction was then kept undisturbed at 1 °C for 24 h. After that, crystal rods were obtained and deposited onto the substrate via a horizontal dipping approach. Then, the product was rinsed with flowing Milli-Q water, ethanol and dried in N$_2$ flow.

### Exfoliation of DY2DP-Por and GDY crystals

0.1 mg of DY2DP-Por or GDY crystals were dispersed into 0.5 ml of water and kept in sealed vials. The vial was sonicated for 1 h in a bath sonication (BANDELIN electronic GmbH, Berlin, Germany) at a fixed nominal power output of 160 W. The temperature of the Milli-Q water in the ultrasonic bath was kept below 30 °C. After sonication, DY2DP-Por or GDY nanoflakes were deposited on a Si/SiO$_2$ wafer and a TEM grid by drop-casting and dried overnight.

### In-situ PM-IRRAS

IRRAS spectra were collected on a Bruker IINVENIO R equipped with a N$_2$ cooled MCT detector and an A511 reflection module (Bruker Optics GmbH, Ettlingen, Germany), placed over a Langmuir-Blodgett (LB) trough setup (RIEGLER KIRSTEIN, Potsdam, Germany). In-situ experiments were performed on the sample LB trough equipped with a Wilhelmy balance. A smaller reference LB trough was placed next to the sample trough for the background measurement. The filling amounts in the reference trough were consistent with the sample one. The spectra were collected using an incidence angle of $\alpha_i$ = 60° with both s- and p-polarized IR light. All acquired spectra were baseline-corrected and also corrected for ambient water vapor and CO$_2$ contributions.

### In-situ grazing incidence X-ray diffraction and fluorescence

The in-situ GIWAXS, GISAXS and XRF measurements were performed at beamline P08 high-resolution diffraction at the 3$^{rd}$ generation synchrotron radiation source PETRA III in DESY, Hamburg, Germany. The beam with a photon energy of 15 keV (120 mA, 480 bunch, high beta) was used. The experiments were performed on the DMAc-H$_2$O surface in a PTFE trough, which was placed in a helium filled enclosure to reduce air scattering and beam damage to the films. For GIWAXS measurements, the Mython2 1 K detector with Soller slits was scanned along the in-plane direction of the DMAc-H$_2$O surface to record the signals for exposure of 10 s at each 2$\theta$

angle. The process involved integrating the individual images horizontally to derive the vertical intensity distribution (denoted as I($Q_z$)) corresponding to each $2\theta$ angle. These 1D spectra were combined to construct a 2D $Q_{xy}$-$Q_z$ intensity maps, which were analyzed using WxDiff. The Eiger2 1 M and Amptek X-123SDD-fast (70 mm$^2$ sensor) detectors were utilized to record the GISAXS images and the XRF spectra, respectively.

## Data availability

The data supporting the findings of this study are available within the article and its Supplementary Information. The crystallographic coordinates for the structure reported in this article are provided in the Source Data file. All data are available from the corresponding author upon request. Experimental procedures and characterizations are available in the Supplementary Information. Source data are provided with this paper.

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

## Acknowledgements

This work was financially supported by the ERC Synergy Grant (2DPolyMembrane, grant no. 101167472), ERC starting grant (FC2DMOF, grant no. 852909), ERC Consolidator Grant (T2DCP), DFG project (2D polyanilines, no. 426572620), GRK2861 (no. 491865171), CRC 1415 (Chemistry of Synthetic Two-Dimensional Materials, no. 417590517), as well as the German Science Council and Center of Advancing Electronics Dresden. R.D. thanks National Natural Science Foundation of China (22272092 & 22472085) and Natural Science Foundation of Shandong Province (ZR2023JQ005). Y.Y. gratefully acknowledges funding from the China Scholarship Council. D.B. A.D. and G.C. thank the Center for Information Services and High Performance Computing (ZIH), TUD Dresden University of Technology, Dresden, Germany for the computational resources. M.P. and T.D.K. were supported by the Center for Advanced Systems Understanding which is financed by Germany's Federal Ministry of Education and Research and by the Saxon state government out of the State budget approved by the Saxon State Parliament. The authors acknowledge the Dresden Center for Nanoanalysis at TUD. The authors thank Dr. P. Formanek for the use of the SEM facility and Dr. Darius Pohl and Dr. Bernd Rellinghaus for the use of the TEM facility in Dresden Center for Nanoanalysis. We thank Dr. Alfonsov Alexey for the EPR measurement in Leibniz-Institut für Festkörper- und Werkstoffforschung Dresden. The authors gratefully acknowledge the computing time provided to them on the high-performance computer Noctua 2 at the NHR Center NHR@Paderborn (PC²). We acknowledge DESY (Hamburg, Germany), a member of the Helmholtz Association HGF, for the provision of experimental facilities. Parts of this research were carried out at PETRA III and we would like to thank Dr. Chen Shen, Rene Kirchhof and Dahdouli Monika for assistance in using photon beamline P08. Beamtime was allocated for proposals I-20230315 and I-20230364. The authors acknowledge the Federal Ministry of Education and Research (BMBF) of the Federal Republic of Germany for financing the Eiger 2 detector via the grant ErUM Pro 05K19FK2 (Murphy), and we thank Professor Olaf Magnussen, P.D. Dr Bridget Murphy from Kiel University, Germany for providing the detector. The research leading to this result has been supported by the project CALIPSOplus under the Grant Agreement 730872 from the EU Framework Programme for Research and Innovation HORIZON 2020.

## Author contributions

Z.W. and X.F. conceived and designed the project. Y.Y. and Z.W. contributed to the synthesis. Y.Y. performed OM, SEM, AFM, NMR, Maldi-TOF, XPS, XRD, ATR-FTIR and UV-vis measurements. Y.Y. performed HRTEM imaging, SAED measurements, and the corresponding analysis. Y.Y. performed the in-situ IRRAS measurement. Y.Y., Z.W., R.D., M.H. and S.M. performed the in-situ GIWAXS, GISAXS, and XRF measurements. C.L. performed the theoretical calculations of the structures. M.P. and T.K. contributed to the energy calculation of the oriented TEPP molecules with different angles. D.B., A.D. and G.C. contributed to the energy simulations of the Glaser coupling reaction. Y.W., T.D., and S.P. contributed to the electrical measurements. Y.Y., Z.W. and X.F. co-wrote the manuscript with contributions from all the authors.

## Funding

## Competing interests

The authors declare no competing interests.
