## [Transparent Peer Review file · Nature Communications]

On-Liquid Surface Synthesis of Diyne-Linked Two-Dimensional Polymer Crystals

Corresponding Author: Professor Xinliang Feng

Version 0:

Reviewer comments:

Reviewer #1

(Remarks to the Author)

The manuscript by Ye Yang et al. reports an on-organic liquid surface synthesis strategy, assisted by a perfluorinated surfactant (PFS) monolayer (O-SMAIS), for the construction of diyne-linked 2D polymer crystals via Glaser coupling on the surface of a DMAc–H₂O mixture. The authors demonstrated that the Glaser coupling on the DMAc–H₂O surface exhibits enhanced reactivity compared to the bulk solution. Through a combination of cutting-edge in-situ spectroscopic (e.g., IRRAS, XRF) and diffraction (e.g., GIXD) techniques, the authors systematically investigated the monomer preorganization and polymerization dynamics by monitoring the stepwise evolution on the DMAc–H₂O surface, including PFS monolayer formation, electrostatically driven adsorption of Cu⁺ ions, and vertical assembly and polymerization of acetylenic monomers. On this basis, micro-scale rod-shaped DY2DP-Por and GDY crystals were obtained, which can be exfoliated into ultrathin 2D nanoflakes. This work meets the general standards of Nature Communications and represents a significant contribution to the fields of interfacial chemistry and materials science. Thus, I recommend the acceptance of this manuscript pending minor revisions as outlined below.

1. As discussed in Line 146, the model reactions were performed to compare the reactivity of Glaser coupling on the surface and in the solution. But the synthesis procedure underlying the model reactions is insufficiently described. The authors are encouraged to provide a more detailed account of the experimental setup and conditions used.
2. While the authors state that in-situ XRF was used to monitor Cu⁺ ion concentration on the DMAc–H₂O surface, the methodology for quantifying this concentration is not clearly described. A more thorough explanation of the quantification process, including calibration and calculation methods, is needed.
3. In Fig. 3b, a slight decrease in the C–F vibrational peak is observed from Step II to Step III. The manuscript does not explain the origin or implication of this change. The authors should clarify the potential chemical or structural transformations associated with this observation.
4. The authors present rod-shaped DY2DP-Por and GDY crystals but do not report size distribution data. Statistical analysis of crystal dimensions (e.g., length and diameter) should be included.
5. The manuscript compares GDY crystals synthesized via O-SMAIS with those produced by other methods. To better highlight the advantages of the O-SMAIS strategy, the authors should provide a comparative table (as supplementary material) summarizing the structural and orientational properties of GDY crystals synthesized via different approaches.
6. The authors mention that the irreversible diyne linkages confer high chemical stability to the 2DP network. However, the manuscript lacks experimental validation of the chemical stability of the resulting DY2DP-Por crystals. It would strengthen the manuscript if the authors could provide data to support this claim.
7. As GDY is a well-established material with notable electronic properties, the manuscript would benefit from additional electrical measurements. Specifically, including the conductivity of the synthesized GDY crystals and comparing it with that of reported GDY would offer valuable insight into their electronic properties and material quality.

Reviewer #2

(Remarks to the Author)

This work presented a method for synthesizing 2D polymer crystals through Glaser coupling reaction on the surface of DMAc-H₂O mixed liquid with the aid of perfluorinated surfactant (PFS) monolayer. Through in-situ spectroscopy and diffraction techniques, the authors found that the PFS monolayer promoted the adsorption of Cu⁺ ions, forming a high-concentration catalyst surface, thereby driving the vertical assembly and coupling reaction of monomers. The synthesized

DY2DP-Por and GDY crystals have definite crystal structures and semiconductor properties. There are still some problems need to be further addressed. The paper can be published in Nat Commun after minor revision.

1. As displayed in Fig. 2d, the authors propose an interfacial reaction mechanism, and it is puzzling that there is oxygen involved in the reaction, and then the Cu ions are not oxidized and always participate in the reaction as Cu⁺.
2. Pyridine appears in all of the reaction schematics (Fig. 2) given by the authors, yet it is not used in the section of "Synthesis of DY2DP-Por crystals.", so please explain exactly why.
3. In Fig. S12a, all the data is compressed into straight lines, which makes it difficult for the reader to get useful information from it, and the authors should adjust the way they graph the data to make it clearer and more readable.
4. The discussion of electronic performance lacks sufficient detail. The authors are requested to provide the formula for the conductivity and details of the calculation procedure.
5. DY2DP-Por crystal with AB stacking mode exhibited high conductivity (0.58 S cm⁻¹), and the authors attributed it to the efficient interlayer electronic coupling. Usually, AA stacking modes have stronger interlayer π -stacking effects compared to AB stacking modes and thus can provide favorable channels for carrier transport, so we hope the authors will give more explanation on the carrier transport in DY2DP-Por.
6. In the section of the conclusion, the authors stated "In addition, the O-SMAIS strategy would offer a promising platform for extending interfacial polymerization to other irreversible coupling reactions, such as Suzuki coupling and Sonogashira coupling,.". Since the authors do not mention the extension experiment in the original main article, it is recommended that this description be deleted.
7. There are also writing errors in the text, please check carefully, such as line 214 "Figs. 3b".

Reviewer #3

(Remarks to the Author)

The manuscript by Wang, Feng and co-workers presents a promising new strategy for the surface-induced growth control of crystalline 2D polymers without relying on dynamic (reversible) polymerization chemistry. The resulting materials demonstrate an unprecedented level of crystallinity and alignment, which could significantly advance synthetic methodologies for C–C linked 2D polymers. There is an impressive level of insights in the polymerization process developed via a combination of spectroscopic/diffraction technique, which raises the methodological standards of the field. I would like to see this paper accepted in Nature Comm. However, some of the analysis/proposed explanations are ambiguous and potentially misleading. Therefore, while the work is novel and significant, a major revision would be required prior to acceptance.

1. The discussion of Cu⁺ adsorption on DMAc:H₂O surface seems misleading. Cu is present within DMAc:H₂O bulk and is accumulated at the interface. If one can talk about any "adsorption", that would be on the surface of PFS. Also, molar concentration cannot characterize the adsorption on surfaces; it gives number of species per unit volume—not per unit area. The authors probably refer to the increase "apparent concentration" in the vicinity of the interface, but this needs to be clear. The details of the XRF measurements, including the expected probing depth, should be discussed.
2. How exactly the MALDI-TOF sampling of the interface was done? Either way, I don't see how these measurements (or the NMR) can prove that the reaction happens at the interface: even if reacted in bulk, the hydrophobic product would likely accumulate at the interface after the reaction.
3. The DFT results (Fig. 2e) suggesting an increased reactivity at the interface are problematic. First, the figure suggests that proton transfer from terminal alkyne to pyridine in solution is exothermic (or endergonic – it's not clear what energy is plotted) by -1.3 eV. It does not make sense to me; the relative pK_a of alkyne (~23) is much higher than that of the protonated pyridine (5.5), suggesting ca. +0.5 eV uphill proton transfer. Second, the transition of copper acetylide to diacetylene is shown to be endothermic by 3.4 eV. If that were the case, the diacetylene-linked dimer would have spontaneously broken in these conditions forming the copper acetylide monomers. These are very complex calculations and I sincerely doubt meaningful results could be obtained for such large systems. I don't think they are critical for the paper, though, and suggest simply removing them.
4. I am not convinced that the feature at ~3260 cm⁻¹ in the FTIR spectra is the terminal alkyne. The latter should be a sharp peak, while the observed hump is more resembling the hydrogen-bonded OH stretch. While the spectra are taken in reference to the pure DMAc:H₂O, minor fluctuation in the structure of water (e.g. ice-like layer at the interface with PFS) could lead to change of intensity of the OH vs that in the reference.
5. Two scattering peaks at Q_z = 0.43 and 0.54 Å⁻¹ (d = 14.6 and 11.6 Å) are attributed to the growing 2D polymer. The manuscript should clarify the crystallographic index of these peaks (100 and 110?). I suggested that the PXRD of the isolated polymer (in Fig.4) is also presented in Q-vector (at least, as a second scale), to show the same lattice is probed in both cases.
6. In GDY synthesis (Fig. 1f), HEB-TMS is used directly without any fluoride source. How does TMS deprotection occur under such conditions? The authors should confirm the removal of the protecting group (e.g., XPS of Si).
7. The manuscript emphasizes the need for a PFS monolayer. A control experiment without PFS (or, better, with SA) with analysis of the product crystallinity seems essential to support this claim.
8. The conductivity discussion may need revision. First, it's misleading to relate the observed thermally activation of the conductivity to the optical band-gap. The activation barrier is only 16 meV, so clearly the charge carrier are NOT created by thermal excitation across the band-gap. The measured conductivity could be a result of inadvertent doping by air. What's does EPR analysis show?
9. In light of the above, I am skeptical of the reported value of 0.5 S/cm; in such materials with E_g >1 eV it would generally require a significant amount of doping. More details on the calculation of the channel geometry is needed. The Fig. 5 shows the crystal covered with as many as 8 contacts. What was the actual channel length of the crystal, BETWEEN individual contacts? The description of the fabrication is unclear. Ti and then Au were deposited onto the crystal and THEN platinum was deposited between the crystal and the gold (how?)

10. The authors attribute the conductivity to interlayer electronic coupling. Is this based on an assumption that the c-axis is the fastest growth direction or the actual mapping of the crystallographic direction was done? This should at least be explained, and, ideally, the direction mapped crystallographically.

Other less critical points:

- The role of surface tension is not entirely clear. There are a number of other parameters which could explain why DMAc:H₂O works while other solvents don't (ability to form PFAS monolayers; hydrophobic effect, etc).
- The GISAXS data (Fig. 3d) is hard to read as it is dominated by the shadow of the beam knife. Consider adding arrows to point to the features of interest.
- The term "oxidative addition" may not be appropriate for the proposed mechanism given the CH bond is broken before the coordination with copper.
- The "horizontal transfer" of the polymer needs a clarification. Is this a Langmuir-Schaeffer type transfer (substrate touching the liquid surface)?
- Abbreviation a.u. is used for both arbitrary units (Figs. 2, 3) and atomic units (hartree, Fig. S16).
- The Fig. S18 supporting the epitaxial relation between the PFS and the polymer is hard to decipher. Please add a side view. Also, this point may need additional discussion in the text. Given the low directionality of the electrostatic interaction, how is the epitaxy achieved, without significantly compromising the intra/intersheet interactions in the polymer?
- The XPS Fig. S25 show a very significant amount of C=O (double the amount of CN). Where is this coming from? I do not understand the deconvolution of N1s. What's the nature of the third component and what's the meaning of pyrrolic N-Cu vs pyridinic N-Cu (if Cu is incorporated in the porphyrine, all 4 nitrogens become identical). Also a survey spectrum showing all observed elements (Cu, F, Si, etc.) is needed.
- The volume of Cu and monomer solutions injected in the petri dish should be specified.

Version 1:

Reviewer comments:

Reviewer #1

(Remarks to the Author)

The authors addressed satisfactorily the issues I raised in the previous round of reviews. Following the changes I can recommend publication of the current manuscript.

Reviewer #2

(Remarks to the Author)

The authors have carefully addressed all the comments and revised the manuscript thoroughly.

Reviewer #3

(Remarks to the Author)

The authors provided a remarkably thorough revision and replied to all of my comments in full. I am pleased to recommend the paper for acceptance.

At the author's discretion (no further review is need), I suggest to revise the discussion of N1s N deconvolution in the SI (P.43, Fig. S25). The labelling of "imine" nitrogen seems confusing as there's no imine group in the structure. Possibly, the authors refer to the pyridinic nitrogen of the uncoordinated porphyrine ring, but that would imply that most of the porphyrine rings are uncoordinated. Also, the ratio of peaks assigned to Cu-porphyrine vs Cu-Py is far from the expected 4:1. In my opinion, the more likely scenario is that the 399 eV peak is from Cu-porphyrine, 401 eV is from Cu-Py (expected at higher BE based on electrostatics) and the 398 eV peaks is the pyrrolic N in the uncoordinated H₂-porphyrine. The author could compare the elemental ratio of Cu vs (deconvoluted) N to better support the assignment.

Point-to-point response to the comments from the reviewers

Response: We appreciate all the reviewers for the constructive and insightful comments, which have greatly contributed to the refinement of this research. Our detailed responses to the reviewers' comments are provided below.

Reviewer #1:

General comment. *The manuscript by Ye Yang et al. reports an on-organic liquid surface synthesis strategy, assisted by a perfluorinated surfactant (PFS) monolayer (O-SMAIS), for the construction of diyne-linked 2D polymer crystals via Glaser coupling on the surface of a DMAc–H₂O mixture. The authors demonstrated that the Glaser coupling on the DMAc–H₂O surface exhibits enhanced reactivity compared to the bulk solution. Through a combination of cutting-edge in-situ spectroscopic (e.g., IRRAS, XRF) and diffraction (e.g., GIXD) techniques, the authors systematically investigated the monomer preorganization and polymerization dynamics by monitoring the stepwise evolution on the DMAc–H₂O surface, including PFS monolayer formation, electrostatically driven adsorption of Cu⁺ ions, and vertical assembly and polymerization of acetylenic monomers. On this basis, micro-scale rod-shaped DY2DP-Por and GDY crystals were obtained, which can be exfoliated into ultrathin 2D nanoflakes. This work meets the general standards of Nature Communications and represents a significant contribution to the fields of interfacial chemistry and materials science. Thus, I recommend the acceptance of this manuscript pending minor revisions as outlined below.*

Response: We appreciate the reviewer's encouraging comments and positive recommendation of our work. Point-by-point responses to all the comments raised have been provided below and changes have been made accordingly.

Comment 1. *As discussed in Line 146, the model reactions were performed to compare the reactivity of Glaser coupling on the surface and in the solution. But the synthesis procedure underlying the model reactions is insufficiently described. The authors are encouraged to provide a more detailed account of the experimental setup and*

conditions used.

Response: We appreciate your insightful feedback. The model reactions (i.e., Model I and Model II) were performed on the DMAc-H₂O surface through the O-SMAIS method. Typically, 40 ml of DMAc-Milli-Q water mixture (v:v = 1:1) was added into a crystallization dish (diameter, 6 cm). Then, 28 μ l of PFS solution (1 mg ml⁻¹ in CHCl₃) was spread on the DMAc-H₂O surface using a micropipette to form a stable and crystalline surfactant monolayer. After 30 min, 1 ml of a mixture of CuCl (0.03 mmol) and pyridine (0.06 mmol) aqueous solution was gently injected into the liquid subphase using a syringe. After 2 h, the reaction was initiated by adding 1 ml of either 10-(4-ethynylphenyl)-5,15,20-triphenylporphyrin (**1**) (4.2 μ mol, in DMAc) for Model I, or phenylacetylene (**3**) (98.0 μ mol, in DMAc) for Model II. The reactions were then kept undisturbed at 1 °C for 24 h, yielding macroscopic organic films on the DMAc-H₂O surface. To obtain the products on the DMAc-H₂O surface, the films were transferred onto the SiO₂/Si substrate via a horizontal dipping approach, followed by vacuum drying at 80 °C (Fig. R1). To isolate the bulk-phase products, 2 ml of the subphase liquid was extracted from the bottom of the crystallization dish using a syringe and dried under vacuum at 80 °C, yielding a solid residue deposited on the inner wall of the vial. The collected materials were subsequently re-dissolved in CHCl₃ and CDCl₃ for matrix-assisted laser desorption/ionization-time-of-flight mass spectrometry (MALDI-TOF MS) and nuclear magnetic resonance (NMR) measurements, respectively.

Fig. R1 | **a**, Photographs of the extracted subphase liquid from the bottom of the crystallization dish before and after vacuum drying. **b**, Schematic illustration of on-liquid surface sampling and bulk-phase sampling. **c**, OM image of the Model I product transferred from the DMAc-H₂O surface onto a SiO₂/Si substrate.

Actions:

- a. We have added the above experiment details in the revised supplementary information on Page 5.

Synthesis procedure of Model I. 40 ml of DMAc-Milli-Q water mixture (v:v = 1:1) was added into a crystallization dish (diameter, 6 cm). Then, 28 μl of PFS solution (1 mg ml^{-1} in CHCl_3) was spread on the DMAc- H_2O surface using a micropipette to form a stable and crystalline surfactant monolayer. After 30 min, 1 ml of a mixture of CuCl (0.03 mmol) and pyridine (0.06 mmol) aqueous solution was gently injected into the liquid subphase using a syringe. After 2 h, the reaction was initiated by adding 1 ml of 10-(4-ethynylphenyl)-5,15,20-triphenylporphyrin (**1**) (4.2 μmol , in DMAc). The reactions were then kept undisturbed at 1 $^\circ\text{C}$ for 24 h, yielding macroscopic organic films on the DMAc- H_2O surface. To obtain the products on the DMAc- H_2O surface, the films were transferred onto the SiO_2/Si substrate via a horizontal dipping approach, followed by vacuum drying at 80 $^\circ\text{C}$ (Supplementary Figs. 9b,c). To isolate the bulk-phase products, 2 ml of the subphase liquid was extracted from the bottom of the crystallization dish using a syringe and dried under vacuum at 80 $^\circ\text{C}$, yielding a solid residue deposited on the inner wall of the vial (Supplementary Figs. 9a,b). The collected materials were subsequently re-dissolved in CHCl_3 for matrix-assisted laser desorption/ionization-time-of-flight mass spectrometry.

Synthesis procedure of Model II. 40 ml of DMAc-Milli-Q water mixture (v:v = 1:1) was added into a crystallization dish (diameter, 6 cm). Then, 28 μl of PFS solution (1 mg ml^{-1} in CHCl_3) was spread on the DMAc- H_2O surface using a micropipette to form a stable and crystalline surfactant monolayer. After 30 min, 1 ml of a mixture of CuCl (0.03 mmol) and pyridine (0.06 mmol) aqueous solution was gently injected into the liquid subphase using a syringe. After 2 h, the reaction was initiated by adding 1 ml of phenylacetylene (**3**) (98.0 μmol in DMAc). The reactions were then kept undisturbed at 1 $^\circ\text{C}$ for 24 h, yielding macroscopic organic films on the DMAc- H_2O surface. To obtain the products on the DMAc- H_2O surface, the films were

transferred onto the SiO₂/Si substrate via a horizontal dipping approach, followed by vacuum drying at 80 °C (Supplementary Figs. 9b,c). To isolate the bulk-phase products, 2 ml of the subphase liquid was extracted from the bottom of the crystallization dish using a syringe and dried under vacuum at 80 °C, yielding a solid residue deposited on the inner wall of the vial (Supplementary Figs. 9a,b). The collected materials were subsequently re-dissolved in CDCl₃ for nuclear magnetic resonance measurement.'

- b. Fig. R1 has been added as Supplementary Fig. 9 in the revised Supplementary Information (Page 16).

Comment 2. *While the authors state that in-situ XRF was used to monitor Cu⁺ ion concentration on the DMAc–H₂O surface, the methodology for quantifying this concentration is not clearly described. A more thorough explanation of the quantification process, including calibration and calculation methods, is needed.*

Response: Thanks for your valuable comments. The concentration of Cu⁺ ions on the DMAc-H₂O surface was quantified using a calibration curve established via in-situ XRF spectroscopy. To construct the calibration, a series of CuCl standard solutions in the DMAc-H₂O mixture with concentrations of 0, 1, 10, and 40 mM were prepared. Then, 50 ml of the prepared CuCl solutions were added into a polytetrafluoroethylene (PTFE) trough with a size of 14 cm × 11 cm, respectively, and the corresponding XRF intensities were recorded under identical measurement conditions. The calibration curve of in-situ XRF for Cu⁺ ions was obtained as shown in Supplementary Fig. 17. The linear characteristic across various concentrations with a slope of $\sim 838.22 \pm 8.83$ mM⁻¹ was observed. This calibration enabled us to determine the concentrations of Cu⁺ ions on the DMAc-H₂O surface from Step I to Step III by converting the recorded XRF signals into absolute concentrations.

Supplementary Fig. 17 | Standard curve of Cu ion solutions measured by in situ XRF.

Actions:

We have added the above discussion in the revised Supplementary Information on Page 25.

‘We quantified the concentration of Cu⁺ ions on the DMAc-H₂O surface using a calibration curve established via in-situ XRF spectroscopy. To construct the calibration, a series of CuCl standard solutions in the DMAc-H₂O mixture with concentrations of 0, 1, 10, and 40 mM were prepared. Then, 50 ml of the prepared CuCl solutions were added into a PTFE trough with a size of 14 cm × 11 cm, respectively, and the corresponding XRF intensities were recorded under identical measurement conditions. The calibration curve of in-situ XRF for Cu⁺ ions was obtained as shown in Supplementary Fig. 17. The linear characteristic across various concentrations with a slope of $\sim 838.22 \pm 8.83 \text{ mM}^{-1}$ was observed. This calibration enabled us to determine the concentrations of Cu⁺ ions on the DMAc-H₂O surface from Step I to Step III by converting the recorded XRF signals into absolute concentrations.’

Comment 3. *In Fig. 3b, a slight decrease in the C–F vibrational peak is observed from Step II to Step III. The manuscript does not explain the origin or implication of this change. The authors should clarify the potential chemical or structural transformations associated with this observation.*

Response: Thanks for the constructive comment. The observed decrease in the C-F vibrational peak intensity likely results from a combination of dipole reorientation (*J. Am. Chem. Soc.* **2014**, 136, 11918-11921), and vibrational mode interference (*Anal. Chem.* **2020**, 92, 8117-8124). On the one hand, in Step III, the accumulated Cu⁺ underneath PFS monolayer guides the adsorption and assembly of TEPP monomers on the DMAc-H₂O surface, a process that might slightly perturb the local packing environment of the fluorocarbon chains in PFS molecules. This conformational change results in a reduced intensity of the characteristic C-F vibrational peak relative to that observed in Step II, even though the number of bonds remains unchanged. On the other hand, the underlying assembly of TEPP monomers—and the subsequent formation of the 2D polymer crystals—contribute additional vibrational signals. The emergence of these new interfacial species, with distinct dipole orientations and dynamic molecular environments, would further diminish the relative intensity of the C-F band by interfering with its coherent signal generation.

Actions:

We have added the above discussion in the revised Supplementary Information on Page 32.

‘The observed decrease in the C-F vibrational peak intensity (Fig. 3b) likely results from a combination of dipole reorientation and vibrational mode interference. On the one hand, in Step III, the accumulated Cu⁺ underneath PFS monolayer guides the adsorption and assembly of TEPP monomers on the DMAc-H₂O surface, a process that might slightly perturb the local packing environment of the fluorocarbon chains in PFS molecules. This conformational change results in a reduced intensity of the characteristic C-F vibrational peak relative to that observed in Step II, even though the number of bonds remains unchanged. On the other hand, the underlying assembly of TEPP monomers—and the subsequent formation of the 2D polymer crystals—contribute additional vibrational signals. The emergence of these new interfacial species, with distinct dipole orientations and dynamic molecular environments, would further diminish the relative intensity of the C-F band by interfering with its coherent

signal generation.’

Comment 4. The authors present rod-shaped DY2DP-Por and GDY crystals but do not report size distribution data. Statistical analysis of crystal dimensions (e.g., length and diameter) should be included.

Response: As suggested, the size distributions of DY2DP-Por and GDY crystals have been statistically analyzed, as shown in Fig. R2. DY2DP-Por crystals exhibit a monomodal size distribution with widths of 0.3-0.7 μm and lengths of 6-14 μm (Fig. R2a-c). In contrast, GDY crystals are larger, with widths of 2-9 μm and lengths of 15-50 μm (Fig. R2d-f). Both DY2DP-Por and GDY are presented as micrometer-scale rod-shaped crystals.

Fig. R2 a, OM image of DY2DP-Por crystals. b, Statistical distribution of the width of DY2DP-Por crystals. c, Statistical distribution of the length of DY2DP-Por crystals. d, OM image of GDY crystals. e, Statistical distribution of the width of GDY crystals. f, Statistical distribution of the length of GDY crystals.

Actions:

- a. We have added the above discussion in the revised Supplementary Information on Page 51.

‘The size distributions of DY2DP-Por and GDY crystals have been statistically analyzed. DY2DP-Por crystals exhibit a monomodal size distribution with widths of 0.3-0.7 μm and lengths of 6-14 μm (Supplementary Fig. 42a-c). In contrast, GDY crystals are larger, with widths of 2-9 μm and lengths of 15-50 μm (Supplementary Fig. 42d-f). Both DY2DP-Por and GDY are presented as micrometer-scale rod-shaped crystals.’

- b. Fig. R2 has been added as Supplementary Fig. 42 in the revised Supplementary Information (Page 51).

Comment 5. *The manuscript compares GDY crystals synthesized via O-SMAIS with those produced by other methods. To better highlight the advantages of the O-SMAIS strategy, the authors should provide a comparative table (as supplementary material) summarizing the structural and orientational properties of GDY crystals synthesized via different approaches.*

Response: Following the reviewer’s suggestion, we have compiled a comparative summary of the domain size, orientation and stacking mode of GDY crystals synthesized via the O-SMAIS method and other reported strategies, including chemical vapor deposition (CVD) methods, explosion methods, solvothermal synthesis, on-metal surface synthesis, liquid-liquid interfacial synthesis and gas-liquid interfacial synthesis (Table R1). Compared to other strategies, the O-SMAIS method offers precise control over monomer alignment and directional growth of GDY crystals, yielding highly oriented GDY crystals with enlarged domain sizes.

Table R1. Summary of the domain size, orientation and stacking mode of the GDY crystals synthesized via the O-SMAIS method and other reported strategies.

Synthetic approaches	Domain size (μm)	Orientation	References
Solvothermal synthesis	~0.002	Random	J. Am. Chem. Soc. 2019 , 141, 10677-10683
CVD method	Amorphous	-	Adv. Mater. 2017 , 29, 1604665
Explosion method	~0.01	Random	Chem. Commun. 2017 , 53, 8074-8077

On-metal surface synthesis	N/A	Face-on	Chem. Commun. 2010 , 46, 3256-3258
Liquid-liquid interfacial synthesis	N/A	Face-on	J. Am. Chem. Soc. 2017 , 139, 3145-3152
Gas-liquid interfacial synthesis	~1.5	Face-on	J. Am. Chem. Soc. 2017 , 139, 3145-3152
O-SMAIS	~30	Edge-on	This work

Actions:

- a. We have added the above discussion in the revised manuscript at Line 310-316 on Page 14.

‘A comparative summary of the domain size, orientation and stacking mode of GDY crystals synthesized via the O-SMAIS method and other reported strategies was compiled, including chemical vapor deposition (CVD) methods, explosion methods, solvothermal synthesis, on-metal surface synthesis, liquid-liquid interfacial synthesis and gas-liquid interfacial synthesis (Supplementary Table 1). Compared to other strategies, the O-SMAIS method offers precise control over monomer alignment and directional growth of GDY crystals, yielding highly oriented GDY crystals with enlarged domain sizes.’

- b. Table R1 has been added as Supplementary Table 1 in the revised Supplementary Information (Page 65).

Comment 6. *The authors mention that the irreversible diyne linkages confer high chemical stability to the 2DP network. However, the manuscript lacks experimental validation of the chemical stability of the resulting DY2DP-Por crystals. It would strengthen the manuscript if the authors could provide data to support this claim.*

Response: Thanks for the constructive comment. To evaluate the chemical stability, DY2DP-Por crystals were treated with 1 M HCl, 1 M NaOH aqueous solutions and N,N-dimethylformamide (DMF) for 7 days. TEM imaging shows that DY2DP-Por retains the integrity of the crystal structure (Fig. R3), while Raman spectra reveal that

the diyne linkage exhibits a consistent adsorption peak at $\sim 2,202\text{ cm}^{-1}$, indicating the preservation of the diyne linkage (Fig. R4). These results demonstrate that the diyne linkage endows the DY2DP-Por crystals with a robust structure, enabling exceptional chemical resistance even in harsh conditions.

Fig. R3 | TEM images of DY2DP-Por crystals treated with DMF, 1M HCl and 1 M NaOH. Insets: FFT images of DY2DP-Por crystals treated with DMF, 1M HCl and 1 M NaOH.

Fig. R4 | Raman spectra of DY2DP-Por crystals treated with DMF, 1M HCl and 1 M NaOH.

Actions:

- a. We have added the discussion in the revised manuscript at Line 289-291 on Page 13.

‘These DY2DP-Por crystals exhibit exceptional chemical resistance even in harsh conditions due to their robust diyne linkage (Supplementary Figs. 39 and 40).’

- b. We have added the above discussion in the revised Supplementary Information on Page 48.

‘To evaluate the chemical stability, DY2DP-Por crystals were treated with N,N-dimethylformamide (DMF), 1 M HCl and 1 M NaOH aqueous solutions for 7 days. TEM imaging shows that DY2DP-Por retains the integrity of the crystal structure (Supplementary Fig. 39), while Raman spectra reveal that the diyne linkage exhibits a consistent adsorption peak at $\sim 2,202\text{ cm}^{-1}$, indicating the preservation of the diyne linkage (Supplementary Fig. 40). These results demonstrate that the diyne linkage endows the DY2DP-Por crystals with a robust structure, enabling exceptional chemical resistance even in harsh conditions.’

- c. Figs. R3 and R4 have been added as Supplementary Figs. 39 and 40 in the revised Supplementary Information (Page 48 and 49).

Comment 7. *As GDY is a well-established material with notable electronic properties, the manuscript would benefit from additional electrical measurements. Specifically, including the conductivity of the synthesized GDY crystals and comparing it with that of reported GDY would offer valuable insight into their electronic properties and material quality.*

Response: We appreciate the reviewer for the valuable comment. To investigate the electronic properties of the synthesized GDY, crystal devices were fabricated for conductivity measurements, as shown in Fig. R5. The linear I - V characteristic observed over the voltage range of -0.4-0.4 V confirms an Ohmic contact between the GDY

crystal and gold (Au)/titanium (Ti)/platinum (Pt) wires (Fig. R6). Variable-temperature conductivity measurements reveal a nonlinear decline in conductivity from 300 to 50 K (Fig. R7a), consistent with semiconducting behavior. Arrhenius fitting reveals an activation energy of 62 meV (Fig. R7b), while the Zhabrodkii plot of $\ln(W)$ versus $\ln(T)$ (with $W = d(\ln\sigma)/d(\ln T)$) (where T is the temperature, and σ is the conductivity) shows a negative slope indicative of hopping transport (Fig. R7c). The room temperature conductivity of GDY crystals reaches 0.36 S cm^{-1} , surpassing most reported GDYs synthesized by other strategies (Table R2). However, we like to highlight that correlating conductivity directly to quality or crystallinity remains challenging and we need to be quite careful for this direct comparison, as variations in stacking mode (e.g., AA, AB and ABC), layer orientation (e.g., edge-on, face-on and random) and sample forms (e.g., film and crystal) of the materials, also significantly influence the electronic transport properties despite the shared GDY structure.

Fig. R5 | SEM image of GDY crystal device with Pt contacts.

Fig. R6 | I - V curve of the GDY crystal device.

Fig. R7 | **a**, Temperature-dependent conductivity of GDY crystal ranging from 300 to 50 K. **b**, The plot of $\ln(\sigma)$ versus the reciprocal of the temperature ($1/T$). **c**, Zabraskii plots of GDY crystal, $\ln(W)$ versus $\ln(T)$, where $W = d(\ln\sigma)/d(\ln T)$.

Table R2. Summary of the conductivity values of GDY at room temperature.

Synthetic approaches	Type	Stacking Modes	σ (S cm ⁻¹)	References
On-metal surface synthesis	Film	ABC	2.52×10^{-6}	Chem. Commun. 2010 , 46, 3256-3258
On-metal surface synthesis	Film	N/A	2.42×10^{-8}	Adv. Mater. 2024 , 36, 2400950
Vapor-liquid-solid synthesis	Nanowire	ABC	19	Dalton Trans. 2012 , 41, 730-733
CVD method	Film	N/A	6.72	Adv. Mater. 2017 , 29, 1604665
O-SMAIS	Crystal	AA	0.36	This work

Actions:

- a. We have added the above discussion in the revised manuscript at Line 325-347 on Page 14 and 15.

‘...The conductivities (σ) of DY2DP-Por and GDY crystals were estimated through Ohm’s Law:

$$\sigma = \frac{1}{\rho} = \frac{L}{RA}$$

where ρ is the resistivity, R is the measured resistance, L is the crystal length between Au/Ti/Pt wires, and A is the cross-sectional area of the crystal. L values were measured to be $\sim 0.33 \mu\text{m}$ for DY2DP-Por and $\sim 12.8 \mu\text{m}$ for GDY (Figs. 5a,d).

Their A values were roughly estimated to be $\sim 0.3 \mu\text{m}^2$ for DY2DP-Por and $\sim 40.7 \mu\text{m}^2$ for GDY. Then, their variable-temperature conductivity curves were obtained as shown in Figs. 5b,e. Notably, the room-temperature σ of DY2DP-Por and GDY crystals reaches 0.58 and 0.36 S cm^{-1} , respectively, exceeding most reported 2D COFs, 2DPs and GDYs synthesized by other strategies (Supplementary Tables 2 and 3). The high σ of DY2DP-Por is likely attributed to pyridine doping during the synthesis. However, for GDY, we like to highlight that correlating conductivity directly to quality or crystallinity remains challenging, as variations in stacking mode (e.g., AA, AB and ABC), layer orientation (e.g., edge-on, face-on and random) and sample forms (e.g., film and crystal) of the materials, also significantly influence the electronic transport properties despite the shared structure. The activation energies (E_a) of DY2DP-Por and GDY crystals were estimated using the Arrhenius equation with a logarithmic form:

$$\ln \sigma = \ln \sigma_0 - \frac{E_a}{k_B} \frac{1}{T}$$

where σ_0 is the pre-exponential conductivity factor, k_B is the Boltzmann's constant, and T is the temperature. E_a of DY2DP-Por and GDY were determined to be $\sim 16 \text{ meV}$ and $\sim 62 \text{ meV}$, respectively, by plotting the linear form of $\ln \sigma$ vs. $1/T$. This activation barrier reflects the effective energy that charge carriers must overcome when traversing the fluctuating energy landscape in DY2DP-Por, and it is not related to the bandgap...'

- b. Figs. R5 and R7 have been added as Figs. 5d-f in the revised manuscript (Line 350, Page 15).

Fig. 5| Electronic property of DY2DP-Por and GDY crystals. a,d, SEM images of DY2DP-Por (a) and GDY (d) crystal device with Pt contacts. b,e, Temperature-dependent conductivities of DY2DP-Por (b) and GDY (e) crystals ranging from 300 to 50 K. Inset: The plot of $\ln(\sigma)$ versus the reciprocal of the temperature ($1/T$). c,f, Zabrodskii plots, $\ln(W)$ versus $\ln(T)$, where $W = d(\ln\sigma)/d(\ln T)$.

- c. Fig. R6 have been added as Supplementary Fig. 53 in the revised Supplementary Information (Page 62)
- d. Table R2 has been added as Supplementary Table 3 in the revised Supplementary Information (Page 67).

Reviewer #2:

General comment. *This work presented a method for synthesizing 2D polymer crystals through Glaser coupling reaction on the surface of DMAc-H₂O mixed liquid with the aid of perfluorinated surfactant (PFS) monolayer. Through in-situ spectroscopy and diffraction techniques, the authors found that the PFS monolayer promoted the adsorption of Cu⁺ ions, forming a high-concentration catalyst surface, thereby driving the vertical assembly and coupling reaction of monomers. The synthesized DY2DP-Por and GDY crystals have definite crystal structures and semiconductor properties. There are still some problems need to be further addressed. The paper can be published in Nat Commun after minor revision.*

Response: We appreciate the reviewer for the positive recommendation of our work and the constructive comments. All the suggestions from the reviewer have been carefully addressed and changes have been made accordingly.

Comment 1. *As displayed in Fig. 2d, the authors propose an interfacial reaction mechanism, and it is puzzling that there is oxygen involved in the reaction, and then the Cu ions are not oxidized and always participate in the reaction as Cu⁺.*

Response: Thanks for the insightful comment regarding the role of oxygen and the oxidation state of copper in the Glaser coupling reaction on the liquid surface. As established in previous studies (*ACS Catal.* **2018**, 8, 1161-1172; *Chem* **2020**, 6, 1933-1951), molecular oxygen is involved as a mild oxidant in the Glaser coupling, which does not permanently oxidize all the Cu⁺ to Cu²⁺ ions. Instead, oxygen transiently oxidizes Cu⁺-acetylide species (**3a**) to form reactive intermediates (**3b**), potentially including Cu²⁺ species (i.e. oxidative coupling mechanisms), which facilitate C-C bond formation (Fig. 2d). Note that the overall catalytic cycle is redox-balanced, in which Cu⁺ ions are regenerated after each coupling event, enabling their continued participation. In addition, nitrogen-donor ligands such as pyridine help stabilize Cu⁺ ions throughout the process, preventing their irreversible oxidation and maintaining catalytic activity.

Fig. 2 | d, Mechanism of the Glaser coupling reaction.

Actions:

We have added the above discussion in the revised Supplementary Information on Page 19.

*‘The Glaser coupling reaction undergoes a typical ‘oxidative coupling mechanism’. Typically, oxygen transiently oxidizes Cu^+ -acetylide species (**3a**) to form reactive intermediates (**3b**), potentially including Cu^{2+} species (i.e. oxidative coupling mechanisms), which facilitate C-C bond formation (Fig. 2d). Note that the overall catalytic cycle is redox-balanced, in which Cu^+ ions are regenerated after each coupling event, enabling their continued participation. In addition, nitrogen-donor ligands such as pyridine help stabilize Cu^+ ions throughout the process, preventing their irreversible oxidation and maintaining catalytic activity.’*

Comment 2. *Pyridine appears in all of the reaction schematics (Fig. 2) given by the authors, yet it is not used in the section of “Synthesis of DY2DP-Por crystals.”, so please explain exactly why.*

Response: We thank the reviewer for pointing out this inconsistency, which results from an oversight in updating the reaction conditions during manuscript preparation. In this study, we systematically explored various synthetic conditions for preparing DY2DP-Por and GDY crystals, including reactions both with and without pyridine. We found that the addition of pyridine effectively accelerates the Glaser coupling and facilitates the formation of large DY2DP-Por and GDY crystals. Thereby, pyridine was used in the final optimized conditions. However, we inadvertently neglected to update the

Methods section to reflect this change. We have now corrected this in the revised manuscript to ensure consistency and clarity.

Actions:

We have revised the Methods section in the revised manuscript at Line 380, Page 16 and Line 389, Page 17.

'Synthesis of DY2DP-Por crystals. ... After 30 min stabilization, 1 ml of CuCl (0.03 mmol) and pyridine (0.06 mmol) aqueous solution was gently injected into the water subphase using a syringe. ...

Synthesis of GDY crystals. ... After 30 min stabilization, 1 ml of CuCl (0.03 mmol) and pyridine (0.06 mmol) aqueous solution was gently injected into the water subphase using a syringe. ...'

Comment 3. *In Fig. S12a, all the data is compressed into straight lines, which makes it difficult for the reader to get useful information from it, and the authors should adjust the way they graph the data to make it clearer and more readable.*

Response: Following the reviewer's suggestion, we have replotted the XRF spectra on a logarithmic scale to improve clarity (Fig. R1). The results indicate that the PFS monolayer promotes electrostatically driven adsorption of Cu⁺ ions, leading to a Cu⁺-rich DMAc-H₂O surface and an ultrahigh intensity of the XRF peak (2.2×10^5). In contrast, without PFS monolayer, Cu⁺ ions show minimal surface accumulation, resulting in a much weaker XRF intensity (1.3×10^3).

Fig. R1 | XRF spectra of the reaction system with and without PFS monolayer after adding CuCl solution for 4 hours on a logarithmic scale.

Actions:

- a. We have added the above discussion in the revised Supplementary Information on Page 27.

‘To enable direct comparison of Cu⁺ ion adsorption on the DMAc-H₂O surface with and without PFS monolayer, the XRF curves after adding CuCl solution for 4 hours were plotted on a logarithmic scale (Supplementary Fig. 19). The PFS monolayer promotes electrostatically driven adsorption of Cu⁺ ions, leading to a Cu⁺-rich DMAc-H₂O surface and an ultrahigh intensity of the XRF peak (2.2×10^5) (Fig. 3c). In contrast, without PFS monolayer, Cu⁺ ions do not aggregate on the DMAc-H₂O surface, resulting in a much weaker XRF intensity (1.3×10^3), which appears almost as a flat line in Supplementary Fig. 18b.’

- b. Fig. R1 has been added as Supplementary Fig. 19 in the revised Supplementary Information (Page 27).

Comment 4. *The discussion of electronic performance lacks sufficient detail. The*

authors are requested to provide the formula for the conductivity and details of the calculation procedure.

Response: Thanks for the constructive comment. We have now included the formula for conductivity and detailed the calculation procedure in the revised manuscript. Conductivity (σ) was calculated using Ohm's Law:

$$\sigma = \frac{1}{\rho} = \frac{L}{RA}$$

where ρ is the resistivity, R is the measured resistance, L is the crystal length between gold (Au)/titanium (Ti)/platinum (Pt) wires, and A is the cross-sectional area of the crystal. We measured L directly from the device geometry and estimated A from the product of the crystal's width and thickness. L values were determined to be $\sim 0.33 \mu\text{m}$ for DY2DP-Por and $\sim 12.8 \mu\text{m}$ for GDY, while A values were roughly estimated to be $\sim 0.3 \mu\text{m}^2$ for DY2DP-Por and $\sim 40.7 \mu\text{m}^2$ for GDY (Figs. R2a,c). Then, their variable-temperature conductivity curves were obtained as shown in Figs. R2b,d. The activation energies (E_a) of DY2DP-Por and GDY crystals were estimated using the Arrhenius equation with a logarithmic form:

$$\ln \sigma = \ln \sigma_0 - \frac{E_a}{k_B} \frac{1}{T}$$

where σ_0 is the pre-exponential conductivity factor, k_B is the Boltzmann's constant, and T is the temperature. E_a of DY2DP-Por and GDY were determined to be $\sim 16 \text{ meV}$ and $\sim 62 \text{ meV}$, respectively, by plotting the linear form of $\ln \sigma$ vs. $1/T$.

Fig. R2| a,c, SEM images of DY2DP-Por (**a**) and GDY (**c**) crystal device with Pt contacts. **b,d,** Temperature-dependent conductivities of DY2DP-Por (**b**) and GDY (**d**) crystals ranging from 300 to 50 K. Inset: The plot of $\ln(\sigma)$ versus the reciprocal of the temperature ($1/T$).

Actions:

a. We have added the above discussion in the revised manuscript at Line 325-345 on Page 14 and 15.

‘...The conductivities (σ) of DY2DP-Por and GDY crystals were estimated through Ohm’s Law:

$$\sigma = \frac{1}{\rho} = \frac{L}{RA}$$

where ρ is the resistivity, R is the measured resistance, L is the crystal length between Au/Ti/Pt wires, and A is the cross-sectional area of the crystal. L values were measured to be $\sim 0.33 \mu\text{m}$ for DY2DP-Por and $\sim 12.8 \mu\text{m}$ for GDY (Figs. 5a,d). Their A values were roughly estimated to be $\sim 0.3 \mu\text{m}^2$ for DY2DP-Por and $\sim 40.7 \mu\text{m}^2$ for GDY. Then, their variable-temperature conductivity curves were obtained as shown in Figs. 5b,e. Notably, the room-temperature σ of DY2DP-Por and GDY

crystals reaches 0.58 and 0.36 S cm^{-1} , respectively, exceeding most reported 2D COFs, 2DPs and GDY synthesized by other strategies (Supplementary Tables 2 and 3). The high σ of DY2DP-Por is likely attributed to pyridine doping during the synthesis. However, for GDY, we like to highlight that correlating conductivity directly to quality or crystallinity remains challenging, as variations in stacking mode (e.g., AA, AB and ABC), layer orientation (e.g., edge-on, face-on and random) and sample forms (e.g., film and crystal) of the materials, also significantly influence the electronic transport properties despite the shared structure. The activation energies (E_a) of DY2DP-Por and GDY crystals were estimated using the Arrhenius equation with a logarithmic form:

$$\ln \sigma = \ln \sigma_0 - \frac{E_a}{k_B} \frac{1}{T}$$

where σ_0 is the pre-exponential conductivity factor, k_B is the Boltzmann's constant, and T is the temperature. E_a of DY2DP-Por and GDY were determined to be $\sim 16 \text{ meV}$ and $\sim 62 \text{ meV}$, respectively, by plotting the linear form of $\ln \sigma$ vs. $1/T$...

b. Fig. 5 has been modified in the revised manuscript (Line 350, Page 15).

Fig. 5 | *Electronic property of DY2DP-Por and GDY crystals. a,d, SEM images of DY2DP-Por (a) and GDY (d) crystal device with Pt contacts. b,e, Temperature-dependent conductivities of DY2DP-Por (b) and GDY (e) crystals ranging from 300 to 50 K. Inset: The plot of $\ln(\sigma)$ versus the reciprocal of the temperature ($1/T$). c,f,*

Zabrodskii plots, $\ln(W)$ versus $\ln(T)$, where $W = d(\ln\sigma)/d(\ln T)$.

Comment 5. DY2DP-Por crystal with AB stacking mode exhibited high conductivity (0.58 S cm^{-1}), and the authors attributed it to the efficient interlayer electronic coupling. Usually, AA stacking modes have stronger interlayer π -stacking effects compared to AB stacking modes and thus can provide favorable channels for carrier transport, so we hope the authors will give more explanation on the carrier transport in DY2DP-Por.

Response: Thanks for this insightful question. We acknowledge that, in principle, AA stacking is often associated with more direct interlayer π -orbital overlaps, which facilitates the charge transport between adjacent layers (Fig. R3). To gain deeper insight into the relationship between stacking mode and charge transport in our case, we carried out comparative DFT calculations of the electronic structures for both AA- and AB-stacked configurations of DY2DP-Por (Fig. R3). As expected, the AA-stacked structure shows pronounced out-of-plane band dispersion due to strong π - π orbital overlap between adjacent layers, indicating low effective mass and high intrinsic mobility along the stacking direction. In contrast, while the interlayer band dispersion in AB stacking is indeed reduced relative to the AA counterpart, the short interlayer spacing ($\sim 0.36 \text{ nm}$) in DY2DP-Por still enables substantial interlayer electronic coupling. This tight spacing promotes orbital proximity sufficient for efficient charge delocalization across layers, contributing to the high conductivity observed experimentally.

Fig. R3| a, Calculated electronic band structure for the AA-stacked DY2DP-Por structure. **b,** Calculated electronic band structure for the AB-stacked DY2DP-Por structure.

In addition to the interlayer electronic coupling, we recently noticed that the

interaction between the polymer backbone and pyridine molecules—used as a catalyst during polymerization—could also modulate the electronic band structure in a way that enhances charge transport. To investigate this, we further calculated the electronic structures of the pyridine-incorporated DY2DP-Por (denoted as DY2DP-Por). As shown in Fig. R4a, compared to pyridine-free DY2DP-Por (denoted as P-DY2DP-Por), DY2DP-Por shows a rigid upshift of both CBM and VBM, along with strong band dispersion along the out-of-plane momentum direction (Fig. R4b), consistent with a p-type semiconducting character and enhanced interlayer charge transport.

To experimentally probe this effect, we employed optical-pump THz-probe (OPTP) spectroscopy—an all-optical, contact-free method for characterizing microscopic charge transport in P-DY2DP-Por and DY2DP-Por. In the measurement, an ultrashort 400 nm laser pulse optically injects charge carriers via above-bandgap excitation, and a time-delayed single-cycle THz pulse (~ 1 ps duration) probes the resulting photoconductivity (*Rev. Mod. Phys.* **2011**, 83, 543-586). The photoconductivity signal normalized by absorbed photon density ($\Delta\sigma/N_{\text{abs}}$) enables a direct comparison of charge carrier mobility (*Nat. Commun.* **2025**, 16, 2219). As shown in Fig. R4c, P-DY2DP-Por exhibits no detectable signal, while DY2DP-Por shows a pronounced signal, several orders of magnitude higher, consistent with the presence of mobile charge carriers and the dispersive valence band structure.

These results suggest that the high conductivity of DY2DP-Por observed in this study likely arises from a synergistic effect of p-type doping, valence band dispersion, and interlayer electronic coupling. Given the uniqueness of this finding and the remarkable photoconductivity response, we plan to further explore this phenomenon in a dedicated study, incorporating extended DFT analysis, OPTP measurements, and device-level investigations.

Fig. R4| a, Calculated electronic band structure for P-DY2DP-Por (without pyridine). **b**, Calculated electronic band structure for DY2DP-Por (with pyridine at their energetically favorable sites). **c**, THz photoconductivity dynamics of P-DY2DP-Por and DY2DP-Por normalized by absorbed photon density.

Actions:

We have revised the manuscript for clarification at Line 335 on Page 15.

‘...The high σ of DY2DP-Por is likely attributed to pyridine doping during the synthesis...’

Comment 6. In the section of the conclusion, the authors stated “In addition, the O-SMAIS strategy would offer a promising platform for extending interfacial polymerization to other irreversible coupling reactions, such as Suzuki coupling and Sonogashira coupling,”. Since the authors do not mention the extension experiment in the original main article, it is recommended that this description be deleted.

Response: Following the reviewer’s suggestion, we have removed the corresponding description to maintain clarity and focus in the manuscript.

Actions:

We have removed the corresponding description in the Conclusion section.

‘...In addition, the O-SMAIS strategy would open the door to synthesizing a wider range of structurally diverse and functional 2DPs with tunable electronic, optical, and mechanical properties.’

Comment 7. *There are also writing errors in the text, please check carefully, such as line 214 “Figs. 3b”.*

Response: Thanks for your valuable comment and we are very sorry for the typesetting errors. We have carefully checked the manuscript and corrected all mistakes. Below are some examples of the corrections made:

Actions:

- a. We have revised the singular form to the plural form in the sentence ‘...1.5 mM, Supplementary Figs. 18-20...’ at Line 215 on Page 10.
- b. We have corrected singular and plural forms in the sentence ‘... (Fig. 3b, 6th curve).’ at Line 221 on Page 10.
- c. We have revised the plural form to the singular form in the sentences ‘...as the monomer (Fig. 1f)...’ at Line 293, Page 13 and ‘...rod-shape GDY crystals (Fig. 4h,...’ at Line 305 on Page 14.
- d. We have standardized the format of conductivity units in Fig. 5b ‘ $S\text{ cm}^{-1}$ ’ at Line 350 on Page 15.
- e. We have corrected the unit of the calculated energy in Supplementary Fig. 23c ‘hartree’ on Page 31.

Reviewer #3:

General comment. *The manuscript by Wang, Feng and co-workers presents a promising new strategy for the surface-induced growth control of crystalline 2D polymers without relying on dynamic (reversible) polymerization chemistry. The resulting materials demonstrate an unprecedented level of crystallinity and alignment, which could significantly advance synthetic methodologies for C–C linked 2D polymers. There is an impressive level of insights in the polymerization process developed via a combination of spectroscopic/diffraction technique, which raises the methodological standards of the field. I would like to see this paper accepted in Nature Comm. However, some of the analysis/proposed explanations are ambiguous and potentially misleading. Therefore, while the work is novel and significant, a major revision would be required prior to acceptance.*

Response: We appreciate the reviewer's support for the significance of our work and the constructive comments provided. Point-by-point responses to all comments raised are provided below, and the manuscript has been revised accordingly.

Comment 1. *The discussion of Cu⁺ adsorption on DMAc:H₂O surface seems misleading. Cu is present within DMAc:H₂O bulk and is accumulated at the interface. If one can talk about any "adsorption", that would be on the surface of PFS.*

Response: Thanks for your valuable comment. We agree that describing Cu⁺ ions as "adsorption" on the DMAc-H₂O surface is misleading. To address this, we have revised the manuscript to clarify that Cu⁺ ions preferentially accumulate underneath the PFS monolayer on the DMAc-H₂O surface, driven by electrostatic interactions between the negatively charged sulfonic acid groups and the Cu⁺ ions.

Actions:

We have corrected the terminology to more accurately reflect the interfacial accumulation in the revised manuscript, including the Fig. 1d at Line 141, Page 6.

Fig. 1| d, Schematic illustration of the synthetic procedure for DY2DPs on the DMAc-H₂O surface

Also, molar concentration cannot characterize the adsorption on surfaces; it gives number of species per unit volume—not per unit area. The authors probably refer to the increase “apparent concentration” in the vicinity of the interface, but this needs to be clear. The details of the XRF measurements, including the expected probing depth, should be discussed.

Response: We thank the reviewer for this constructive comment. Indeed, the measured concentration of Cu⁺ ions on the DMAc-H₂O surface refers to the ‘apparent concentration’. In our study, the in-situ XRF measurement was performed under the total reflection condition (*J. Phys.: Conf. Ser.* **2022**, 2380, 012047), which limits the X-ray penetration depth to ~6 nm. This shallow probing depth ensures that the detected fluorescence signal predominantly arises from ions localized at or near the interface, minimizing contributions from the bulk solution (*J. Appl. Phys.* **2009**, 105, 084911). The XRF signal was collected using an energy-dispersive detector (Amptek X-123SDD, AMETEK Inc., USA) positioned within the experimental housing at an angle of 3° above the horizontal plane and 90° relative to the incident beam direction (Fig. R1). This configuration was chosen to effectively minimize background contributions from Compton and elastic scattering. The X-ray beam used for fluorescence measurements features a spot size of 70 μm × 1,000 μm.

For quantitative analysis, the concentration of Cu⁺ ions on the DMAc-H₂O surface was quantified using a calibration curve established via in-situ XRF spectroscopy. To construct the calibration, a series of CuCl standard solutions in the DMAc-H₂O mixture

with concentrations of 0, 1, 10, and 40 mM were prepared. Then, 50 ml of the prepared CuCl solutions were added into a polytetrafluoroethylene (PTFE) trough with a size of 14 cm × 11 cm, respectively, and the corresponding XRF intensities were recorded under identical measurement conditions. The calibration curve of in-situ XRF for Cu⁺ ions was obtained as shown in Supplementary Fig. 17. The linear characteristic across various concentrations with a slope of $\sim 838.22 \pm 8.83 \text{ mM}^{-1}$ was observed. This calibration enabled us to determine the concentrations of Cu⁺ ions on the DMAc-H₂O surface from Step I to Step III by converting the recorded XRF signals into absolute concentrations.

Fig. R1. Schematic illustration of the in-situ XRF measurement.

Supplementary Fig. 17 Standard curve of Cu ion solutions measured by in situ XRF.

Actions:

- a. We have revised the Methods section in the revised Supplementary Information on

Page 7.

‘In-situ XRF measurement. The in-situ XRF measurement was performed under the total reflection condition, which limits the X-ray penetration depth to ~6 nm. This shallow probing depth ensures that the detected fluorescence signal predominantly arises from ions localized at or near the interface, minimizing contributions from the bulk solution. The XRF signal was collected using an energy-dispersive detector (Amptek X-123SDD, AMETEK Inc., USA) positioned within the experimental housing at an angle of 3° above the horizontal plane and 90° relative to the incident beam direction (Supplementary Fig. 15). This configuration was chosen to effectively minimize background contributions from Compton and elastic scattering. The X-ray beam used for fluorescence measurements features a spot size of 70 μm × 1,000 μm.’

- b. We have added the above discussion in the revised Supplementary Information on Page 26.

‘We quantified the concentration of Cu⁺ ions on the DMAc-H₂O surface using a calibration curve established via in-situ XRF spectroscopy. To construct the calibration, a series of CuCl standard solutions in the DMAc-H₂O mixture with concentrations of 0, 1, 10, and 40 mM were prepared. Then, 50 ml of the prepared CuCl solutions were added into a polytetrafluoroethylene (PTFE) trough with a size of 14 cm × 11 cm, respectively, and the corresponding XRF intensities were recorded under identical measurement conditions. The calibration curve of in-situ XRF for Cu⁺ ions was obtained as shown in Supplementary Fig. 11. The linear characteristic across various concentrations with a slope of ~838.22 ± 8.83 mM⁻¹ was observed. This calibration enabled us to determine the concentrations of Cu⁺ ions on the DMAc-H₂O surface from Step I to Step III by converting the recorded XRF signals into absolute concentrations.’

- c. Fig. R1 has been added as Supplementary Fig. 15 in the revised Supplementary Information (Page 23).

Comment 2. *How exactly the MALDI-TOF sampling of the interface was done? Either*

way, I don't see how these measurements (or the NMR) can prove that the reaction happens at the interface: even if reacted in bulk, the hydrophobic product would likely accumulate at the interface after the reaction.

Response: Thanks for raising this question. In our study, to differentiate between bulk and on-liquid surface reactivity, we employed two separate sampling strategies. To obtain the products on the DMAc-H₂O surface, the films were transferred onto the SiO₂/Si substrate via a horizontal dipping approach, followed by vacuum drying at 80 °C (Figs. R2b,c). To isolate the bulk-phase products, 2 ml of the subphase liquid was extracted from the bottom of the crystallization dish using a syringe and dried under vacuum at 80 °C, yielding a solid residue deposited on the inner wall of the vial (Figs. R2a,b). The collected materials were subsequently re-dissolved in CHCl₃ and CDCl₃ for matrix-assisted laser desorption/ionization-time-of-flight mass spectrometry (MALDI-TOF MS) and nuclear magnetic resonance (NMR) measurements, respectively.

Fig. R2| a, Digital images of the extracted subphase liquid from the bottom of the crystallization dish before and after drying. **b**, Schematic diagram of on-liquid surface sampling and bulk-phase sampling. **c**, OM image of the model I product transferred from the DMAc-H₂O surface onto a SiO₂/Si substrate.

We agree that the hydrophobic nature of the product formed in the liquid subphase might lead to its interfacial accumulation on the liquid surface. However, this presupposes that the reaction proceeds efficiently in the liquid subphase. Typically, the hydrophobic precipitation process results in the majority of the product settling at the bottom of the solution, while a small portion may remain suspended or float on the surface. Note that the MS analysis of the sample collected from the bottom of the

solution revealed the presence of only the unreacted monomer in the solution phase, with no detectable signal corresponding to the product (Fig. 2b). Compared to the MS results of surface sampling, we conclude that the model reaction scarcely proceeds in the solution phase, thereby excluding the possibility that the product accumulates on the surface through migration from the solution. This is similar to our previous studies of the on-water surface model reactions (*Nat. Synth.* **2022**, 1, 69-76; *Angew. Chem. Int. Ed.* **2024**, 63, e202316299).

To further support the surface product formation is not a result of hydrophobic accumulation of products from the subphase, we performed the identical model reaction in the absence of PFS monolayer on the DMAc-H₂O surface. After 24 hours of reactions, we conducted on-liquid surface sampling and attempted to collect any compounds formed on the DMAc-H₂O surface. As shown in Fig R3, no detectable product was observed on the substrate, indicating that no product formed in the subphase and subsequently migrated to the liquid surface via hydrophobic effects. These findings verify that the reactivity on the DMAc-H₂O surface is decoupled from that in the bulk solution, and that the observed product by MS arises only from the on-liquid surface reaction.

Fig. R3 | OM images indicate no product formed on the DMAc-H₂O surface in the absence of PFS monolayer.

Actions:

- a. We have adjusted the phrasing of the discussion in the revised manuscript at Line 155-157, on Page 7.

‘...This suggests that the model reaction scarcely proceeds in the solution phase

(Supplementary Figs. 9 and 10)...

- b. We have added the above discussion in the revised Supplementary Information on Page 16.

'The formation of an organic thin film was observed on the DMAc-H₂O surface after the model reaction. To differentiate between bulk and on-liquid surface reactivity, we employed two separate sampling strategies. To obtain the products on the DMAc-H₂O surface, the films were transferred onto the SiO₂/Si substrate via a horizontal dipping approach, followed by vacuum drying at 80 °C (Supplementary Figs. 9b,c). To isolate the bulk-phase products, 2 ml of the subphase liquid was extracted from the bottom of the crystallization dish using a syringe and dried under vacuum at 80 °C, yielding a solid residue deposited on the inner wall of the vial (Supplementary Figs. 9a,b). The collected materials were subsequently re-dissolved in CHCl₃ and CDCl₃ for matrix-assisted laser desorption/ionization-time-of-flight mass spectrometry (MALDI-TOF MS) and nuclear magnetic resonance (NMR) measurements, respectively.

To further support the surface product formation is not a result of hydrophobic accumulation of products from the subphase, we performed the identical model reaction in the absence of PFS monolayer on the DMAc-H₂O surface. After 24 hours of reactions, we conducted on-liquid surface sampling and attempted to collect any compounds formed on the DMAc-H₂O surface. As shown in Supplementary Fig. 10, no detectable product was observed on the substrate, indicating that no product formed in the subphase and subsequently migrated to the liquid surface via hydrophobic effects. These findings verify that the reactivity on the DMAc-H₂O surface is decoupled from that in the bulk solution, and that the observed product by MS arises only from the on-liquid surface reaction.'

- c. Figs. R2 and R3 has been added as Supplementary Figs. 9 and 10 in the revised Supplementary Information (Page 16 and 17).

Comment 3. *The DFT results (Fig. 2e) suggesting an increased reactivity at the interface are problematic. First, the figure suggests that proton transfer from terminal*

alkyne to pyridine in solution is exothermic (or endergonic – it's not clear what energy is plotted) by -1.3 eV. It does not make sense to me; the relative pK_a of alkyne (~ 23) is much higher than that of the protonated pyridine (5.5), suggesting ca. $+0.5$ eV uphill proton transfer.

Response: Thanks for the constructive comment. In our initial mechanistic study, we considered the possibility that the Glaser coupling reaction might proceed via the Eglinton coupling mechanism (*J. Chem.* **2015**, 1, 430358), which involves a discrete deprotonated terminal alkyne intermediate. This assumption led us to include a simplified DFT model showing proton transfer from the terminal alkyne to pyridine. However, upon further evaluation of the literature and reaction conditions (Fig. R4), we recognize that the mechanism in our system, catalyzed by Cu^+ ions, is more accurately described by the classical Glaser coupling mechanism (*Chem* **2020**, 6, 1933-1951; *Eur. J. Org. Chem.* **2013**, 4, 701-711). In this case, terminal alkyne deprotonation does not occur via direct proton transfer to pyridine in bulk solution. Rather, the deprotonation is facilitated by the coordination of Cu^+ ion with the terminal alkyne, and the presence of pyridine helps stabilize the resulting Cu-acetylide complex. To reflect the energetics or mechanistic pathway correctly, we have removed the deprotonated terminal alkyne intermediate in Fig. 2d and its corresponding description.

Fig. R4| Proposed mechanism for Eglinton coupling and classical Glaser coupling (*Eur. J. Org. Chem.* **2013**, 701-711).

Fig. 2 | c, Schematic of Model II reactions. Compounds **3** yield **4**. d,e, Mechanism (d) and its calculated energy profiles (e) of the Glaser coupling reaction. The energy barriers are with respect to **3**.

Actions:

- The discussion on the deprotonation of terminal alkyne has been modified.
- Figs. 2d,e have been modified in the revised manuscript at Line 176 on Page 8.
- Supplementary Fig. 13 have been modified in the revised Supplementary Information on Page 21.

Second, the transition of copper acetylide to diacetylene is shown to be endothermic by 3.4 eV. If that were the case, the diacetylene-linked dimer would have spontaneously broken in these conditions forming the copper acetylide monomers. These are very complex calculations and I sincerely doubt meaningful results could be obtained for such large systems. I don't think they are critical for the paper, though, and suggest simply removing them.

Response: We appreciate the valuable comment. We would like to highlight that the calculations are intended to compare the reactivity differences between the bulk solution phase and the liquid surface, rather than precise thermodynamic values. The calculated high endothermic barrier of 3.38 eV for the transformation of the Cu⁺-acetylene intermediate into the diacetylene product underscores the substantial energy barrier associated with this process in the DMAc-H₂O solution. This result is consistent

with the MALDI-TOF MS data (Fig. 2b), where only unreacted monomer was detected in the bulk DMAc-H₂O solution, indicating that the formation of the diyne bond is unfavorable.

Importantly, this high energy difference does not imply that the diyne bond would have spontaneously broken in the DMAc-H₂O solution, as the calculation was based solely on two representative intermediate states to estimate the associated energy barriers. Additional intermediate states, such as the dicopper²⁺-diacetylide complex with various molecular conformations or even the dicopper³⁺-diacetylide intermediate, likely exist between the diacetylene-linked dimer and Cu⁺-acetylene complex, and these may involve higher activation energy barriers (*Tetrahedron* **2002**, 58, 6741-6747). While we agree that the calculation is highly complex, we believe the current results offer valuable insights into the qualitative difference in reactivity between the liquid surface and bulk solution. To ensure our conclusions remain well-balanced, we have revised the Supplementary Information to present the computational findings more cautiously.

Actions:

We have added the above discussion in the revised Supplementary Information on Page 21.

‘The calculated high endothermic barrier of 3.38 eV for the transformation of the Cu⁺-acetylene intermediate into the diacetylene product underscores the substantial energy barrier associated with this process in the DMAc-H₂O solution. This result is consistent with the MALDI-TOF MS data (Fig. 2b), where only unreacted monomer was detected in the bulk DMAc-H₂O solution, indicating that the formation of the diyne bond is unfavorable.

Importantly, this high energy difference does not imply that the diyne bond would have spontaneously broken in the DMAc-H₂O solution, as the calculation was based solely on two representative intermediate states to estimate the associated energy barriers. Additional intermediate states, such as the dicopper²⁺-diacetylide complex with various molecular conformations or even the dicopper³⁺-diacetylide intermediate,

likely exist between the diacetylene-linked dimer and Cu^+ -acetylene complex, and these may involve higher activation energy barriers.'

Comment 4. *I am not convinced that the feature at $\sim 3260\text{ cm}^{-1}$ in the FTIR spectra is the terminal alkyne. The latter should be a sharp peak, while the observed hump is more resembling the hydrogen-bonded OH stretch. While the spectra are taken in reference to the pure DMAc:H₂O, minor fluctuation in the structure of water (e.g, ice-like layer at the interface with PFS) could lead to change of intensity of the OH vs that in the reference.*

Response: We thank the reviewer for this insightful comment. To clarify the origin of the broad peak at $3,260\text{ cm}^{-1}$, either from terminal alkyne or from OH stretch, we further analyzed the in-situ IRRAS data in the wavelength range from $4,000$ to $2,500\text{ cm}^{-1}$. As shown in Fig. R5, a broad absorption peak at $\sim 3,680\text{ cm}^{-1}$, corresponding to the -OH stretching, was observed in the pure DMAc-H₂O system. Upon addition of the PFS monolayer, this peak undergoes significant decreases in intensity, due to the surface crowding (*Angew. Chem. Int. Ed.* **2021**, 60, 25143) and water structure fluctuation (*J. Phys. Chem. Lett.* **2021**, 12, 218-223).

Regarding the new absorption peak at $3,264\text{ cm}^{-1}$, we note that it emerges exclusively following the addition of the TEPP monomer. Its position, associated with its attenuation during the 2D polymerization strongly suggest assignment to the terminal alkyne $\text{C}\equiv\text{C-H}$ stretching. While terminal alkynes typically produce a sharp, well-defined peak in solid-state FTIR (as also observed in our ex-situ FTIR measurement, Supplementary Fig. 32b), the peak on the DMAc-H₂O surface is broadened. This broadening likely arises from polar solvent interactions and heterogeneous local environments around the terminal alkyne groups, which perturb the vibrational dynamics, leading to noticeable band broadening. Such solvent-induced broadening effects have also been well-documented in the literature (*J. Phys. Chem. A* **2009**, 113, 1760-1769; *J. Electrochem. Soc.* **2014**, 161, H738). To avoid potential misleading and confusion, we have clarified this point in the revised manuscript.

Fig. R5 | Time evolution of the PM-IRRAS spectra with an angle of incidence $\alpha_i = 60^\circ$ in p-polarization recorded on the DMAc-H₂O surface from Step I to Step III.

Actions:

- a. We have added the discussion in the revised manuscript at Line 189-192 on Page 8 and 9.

‘...In-situ PM-IRRAS spectra show a broad absorption peak at $\sim 3,680\text{ cm}^{-1}$, corresponding to the -OH stretching, was observed in the pure DMAc-H₂O system (Supplementary Fig. 16). Upon addition of the PFS monolayer, this peak undergoes significant decreases in intensity, due to the surface crowding and water structure fluctuation...’

- b. We have added the above discussion in the revised Supplementary Information on Page 24.

‘Regarding the new absorption peak at $3,264\text{ cm}^{-1}$, we note that it emerges exclusively following the addition of the TEPP monomer. Its position, associated with its attenuation during the 2D polymerization strongly suggest assignment to the terminal alkyne $\text{C}\equiv\text{C-H}$ stretching. While terminal alkynes typically produce a

sharp, well-defined peak in solid-state FTIR (as also observed in our ex-situ FTIR measurement, Supplementary Fig. 32b), the peak on the DMAc-H₂O surface is broadened. This broadening likely arises from polar solvent interactions and heterogeneous local environments around the terminal alkyne groups, which perturb the vibrational dynamics, leading to noticeable band broadening.’

c. Fig. R5 has been added as Supplementary Fig. 16 in the revised Supplementary Information (Page 24).

Comment 5. Two scattering peaks at $Q_z = 0.43$ and 0.54 \AA^{-1} ($d = 14.6$ and 11.6 \AA) are attributed to the growing 2D polymer. The manuscript should clarify the crystallographic index of these peaks (100 and 110?). I suggested that the PXRD of the isolated polymer (in Fig.4) is also presented in Q -vector (at least, as a second scale), to show the same lattice is probed in both cases.

Response: Thanks for the constructive comment. The GISAXS peaks at $Q_z = 0.43$ and 0.54 \AA^{-1} are assigned to the (110) and (200) plane of the DY2DP-Por crystal, respectively. These assignments are consistent with the first two peaks in the XRD pattern. Following the reviewer’s suggestion, we have converted the XRD data from 2θ value to Q vector, now shown in Fig. R6. Moreover, the crystallographic index of these peaks has been marked in Figs. 3d,e in the revised manuscript.

Fig. R6| a,b, XRD patterns of DY2DP-Por (a) and GDY (b) crystals and the simulations. The 2θ values were converted to Q vector follows the equation: $Q = 4\pi\sin(\theta)/\lambda$. Insets: the simulated structure from the top view.

Fig. 3| d, Representative in-situ GISAXS patterns for Step I-III measured on the DMAC-H₂O surface. **e**, Time evolution of the out-of-plane GISAXS projections (near $Q_{xy} = 0$, Q represents scattering vector) during Step I-III. **f**, Calculated concentration of Cu⁺ on the DMAC-H₂O surface through the peak at the photon energy of 8.04 eV in the in-situ XRF spectra. **g**, Time-dependent PM-IRRAS integral areas of the vibrational peak of C≡C-H (3264 cm⁻¹) for p-polarization as well as the calculated crystallite size of DY2DP-Por using the full width at half maximum of the Gaussian fit of the peak at $Q_z = 0.54 \text{ \AA}^{-1}$ in the in-situ GISAXS spectra.

Actions:

a. We have added the corresponding Q vector in the revised manuscript at Line 276, Page 13 and Line 300, Page 13.

‘...The XRD patterns exhibit Bragg peaks at $Q = 0.42, 0.54, 0.63, 0.88, 1.03,$ and 1.19 \AA^{-1} ($2\theta = 5.91^\circ, 7.69^\circ, 8.92^\circ, 12.45^\circ, 14.56^\circ,$ and 16.79°), corresponding to the (110), (200), (020), (021), (212) and (141) planes, respectively...’

‘...The sharp Bragg peaks at $Q = 0.77, 0.81, 1.37, 1.54,$ and 1.88 \AA^{-1} ($2\theta = 10.86^\circ,$

11.35°, 19.42°, 21.84°, and 26.62°) were probed in the XRD spectra, corresponding to the (100), (010), (110), (200) and (001) planes, respectively (Fig. 4g)...

b. Figs. 4b,g in the revised manuscript has been modified at Line 262 on Page 12.

Comment 6. In GDY synthesis (Fig. 1f), HEB-TMS is used directly without any fluoride source. How does TMS deprotection occur under such conditions? The authors should confirm the removal of the protecting group (e.g., XPS of Si).

Response: Thanks for pointing this out. Under our reaction conditions, the TMS deprotection is facilitated by the presence of excess CuCl, which promotes the formation of Cu⁺-acetylene complex and hexamethyldisiloxane, as previously reported (*Angew. Chem. Int. Ed.* **2022**, 61, e202210242; *Chem. Sci.* **2021**,12, 12661-12666; *Inorg. Chem.* **2024**, 63, 21679-21686). To experimentally confirm the removal of TMS group, we further conducted XPS and Raman measurements on the resulting GDY crystals (Figs. R7 and R8). The XPS survey spectra shows no detectable silicon signal, demonstrating the efficient deprotection of TMS during the 2D polymerization. Moreover, the Raman peak of HEB-TMS at 2,897 cm⁻¹, corresponding to the -CH₃ in TMS, disappears completely in the GDY crystals. These results collectively confirm the removal of TMS group during GDY formation.

Fig. R7 | XPS survey scan of GDY crystals. Inset: High-resolution XPS spectra in the binding energy region of 0-200 eV.

Fig. R8 | Raman spectra of HEB-TMS (top) and GDY crystals (bottom).

Actions:

- a. We have added the above discussion in the revised Supplementary Information on Page 53.

‘Under our reaction conditions, the TMS deprotection is facilitated by the presence of excess CuCl, which promotes the formation of Cu⁺-acetylene complex and hexamethyldisiloxane. To experimentally confirm the removal of TMS group, we further conducted XPS and Raman measurements on the resulting GDY crystals (Supplementary Figs. 44 and 45). The XPS survey spectra shows no detectable silicon signal, demonstrating the efficient deprotection of TMS during the 2D polymerization. Moreover, the Raman peak of HEB-TMS at 2897 cm⁻¹, corresponding to the -CH₃ in TMS, disappears completely in the GDY crystals. These results collectively confirm the removal of TMS group during GDY formation.’

- b. Figs. R7 and R8 have been added as Supplementary Figs. 44 and 45 in the revised Supplementary Information (Page 53 and 54).

Comment 7. The manuscript emphasizes the need for a PFS monolayer. A control experiment without PFS (or, better, with SA) with analysis of the product crystallinity seems essential to support this claim.

Response: Following the reviewer's suggestion, we have conducted control experiments in the absence of the PFS monolayer and in the presence of stearic acid (SA) to evaluate their role in the 2D polymerization process on the DMAc-H₂O surface. Without the stable PFS monolayer on the DMAc-H₂O surface, the in situ XRF spectra reveal that the Cu⁺ ion concentration increases negligibly from 0 to 1.49 mM in Step II (vs. from 0 to 265 mM in the presence of PFS), indicating a homogenous ion diffusion process instead of the accumulation process driven by electrostatic interaction (Supplementary Fig. 18). Upon the addition of TEPP monomers, no detectable GIWAXS and GISAXS signals were observed in either the in-plane or out-of-plane directions, indicating the absence of oriented monomer assembly or growth of DY2DP-Por crystals (Supplementary Figs. 28-30). Even after 24 hours, no product was formed on the DMAc-H₂O surface (Fig. R9). In the system using SA, we note that it cannot form the stable monolayer on the DMAc-H₂O surface, as confirmed by the isotherm measurements of surface pressure (Fig. 1b). After 24-hour polymerization, we did not observe any product formed on the DMAc-H₂O surface through the OM imaging (Fig. R10). These results underscore the crucial role of the PFS monolayer in directing the accumulation of Cu⁺ and oriented growth of DY2DP-Por crystals.

Supplementary Fig. 18| a,b, Time evolution of the XRF spectra recorded on the DMAc-H₂O surface (a) with and (b) without the PFS monolayer. c, In situ XRF spectra

of the DMAc-H₂O surface before and after adding CuCl without the PFS monolayer.

Supplementary Fig. 28 | Time evolution of the GIWAXS patterns of 2D polymerization recorded on the DMAc-H₂O surface without the PFS monolayer.

Supplementary Fig. 29 | Time evolution of the GISAXS patterns of 2D polymerization recorded on the DMAc-H₂O surface without the PFS monolayer.

Supplementary Fig. 30| a, Time evolution of the in-plane GIWAXS projections (near $Q_z = 0$, Q represents scattering vector) of 2D polymerization recorded on the DMAC-H₂O surface without the PFS monolayer. **b**, Time evolution of the out-of-plane GISAXS projections (near $Q_{xy} = 0$) of 2D polymerization recorded on the DMAC-H₂O surface without the PFS monolayer.

The system without the PFS monolayer

Fig. R9| OM images of the products transferred from the DMAC-H₂O surface in the system without the PFS monolayer.

Fig.1| b, Surface pressures of the DMAC-H₂O surface after adding SA and PFS,

respectively.

The system using SA

Fig. R10 | OM images of the products transferred from the DMAc-H₂O surface in the system using SA.

Actions:

- a. We have added the above discussion in the revised Supplementary Information on Page 26.

‘We have conducted control experiments in the absence of the PFS monolayer and in the presence of stearic acid (SA) to evaluate their role in the 2D polymerization process on the DMAc-H₂O surface. Without the stable PFS monolayer on the DMAc-H₂O surface, the in situ XRF spectra reveal that the Cu⁺ ion concentration increases negligibly from 0 to 1.49 mM in Step II, indicating a homogenous ion diffusion process instead of the accumulation process driven by electrostatic interaction (Supplementary Figs. 18-20). Upon the addition of TEPP monomers, no detectable GIWAXS and GISAXS signals were observed in either the in-plane or out-of-plane directions, indicating the absence of oriented monomer assembly or growth of DY2DP-Por crystals (Supplementary Figs. 28-30). Even after 24 hours, no product was formed on the DMAc-H₂O surface (Supplementary Fig. 26). In the system using SA, we note that it cannot form the stable monolayer on the DMAc-H₂O surface, as confirmed by the isotherm measurements of surface pressure (Fig. 1b). After 24-hour polymerization, we did not observe any product formed on the DMAc-H₂O surface through the OM imaging (Supplementary Fig. 27). These results underscore the crucial role of the PFS monolayer in directing the

accumulation of Cu⁺ and oriented growth of DY2DP-Por crystals.'

- b. Figs. R9 and R10 have been added as Supplementary Figs. 26 and 27 in the revised Supplementary Information (Page 35 and 36).

Comment 8. *The conductivity discussion may need revision. First, it's misleading to relate the observed thermal activation of the conductivity to the optical band-gap. The activation barrier is only 16 meV, so clearly the charge carriers are NOT created by thermal excitation across the band-gap. The measured conductivity could be a result of inadvertent doping by air. What's does EPR analysis show?*

Response: Thanks for the constructive comment. Indeed, the activation barrier reflects the effective energy that charge carriers must overcome when traversing the fluctuating energy landscape in DY2DP-Por, and it is not related to the bandgap. We have revised the manuscript to avoid misleading.

To investigate whether the measured conductivity arises from unintentional air doping, we conducted temperature-dependent electron paramagnetic resonance (EPR) on DY2DP-Por at 5, 25, 50 and 100 K (Fig. R11). The spectra reveal strong, well-defined signals corresponding to Cu²⁺ ions coordinated within the porphyrin macrocycle of DY2DP-Por. Importantly, no additional EPR features attributable to organic radicals or air-induced dopants were detected across the temperature range studied, thereby ruling out air doping as a contributing factor to the measured electronic conductivity.

Fig. R11 | Temperature-dependent EPR spectra of DY2DP-Por at 5, 25, 50 and 100 K.

Actions:

- a. We have added the above clarification in the revised manuscript at Line 345-347 on Page 15.

‘This activation barrier reflects the effective energy that charge carriers must overcome when traversing the fluctuating energy landscape in DY2DP-Por, and it is not related to the bandgap.’

- b. We have added the above discussion in the revised Supplementary Information on Page 64.

‘To investigate whether the measured conductivity arises from unintentional air doping, we conducted temperature-dependent electron paramagnetic resonance (EPR) on DY2DP-Por at 5, 25, 50 and 100 K (Supplementary Fig. 55). The spectra reveal strong, well-defined signals corresponding to Cu^{2+} ions coordinated within the porphyrin macrocycle of DY2DP-Por. Importantly, no additional EPR features attributable to organic radicals or air-induced dopants were detected across the temperature range studied, thereby ruling out air doping as a contributing factor to the measured electronic conductivity.’

- c. Fig. R11 has been added as Supplementary Fig. 55 in the revised Supplementary Information (Page 64).

Comment 9. *In light of the above, I am skeptical of the reported value of 0.5 S/cm; in such materials with $E_g > 1$ eV it would generally require a significant amount of doping.*

Response: We thank the reviewer for the valuable comment. Our latest findings suggest that the high conductivity observed in DY2DP-Por may stem from subtle changes in its electronic band structure induced by molecular interactions with pyridine—the catalyst used during polymerization. To investigate this, we performed density functional theory (DFT) calculations comparing the pyridine-free DY2DP-Por (denoted as P-DY2DP-Por) and a model incorporating pyridine molecules at their energetically favorable sites (denoted as DY2DP-Por). As shown in Fig. R12a, P-DY2DP-Por exhibits flat conduction band minimum (CBM) and valence band maximum (VBM), with the Fermi level lying mid-gap, indicative of insulating behavior. In contrast, DY2DP-Por shows a rigid upshift of both CBM and VBM, along with strong band dispersion along the out-of-plane momentum direction (Fig. R12b), consistent with a p-type semiconducting character and enhanced interlayer charge transport.

To experimentally probe this effect, we employed optical-pump THz-probe (OPTP) spectroscopy—an all-optical, contact-free method for characterizing microscopic charge transport in P-DY2DP-Por and DY2DP-Por. In the measurement, an ultrashort 400 nm laser pulse optically injects charge carriers via above-bandgap excitation, and a time-delayed single-cycle THz pulse (~1 ps duration) probes the resulting photoconductivity (*Rev. Mod. Phys.* **2011**, 83, 543-586). The photoconductivity signal normalized by absorbed photon density ($\Delta\sigma/N_{\text{abs}}$) enables a direct comparison of charge carrier mobility (*Nat. Commun.* **2025**, 16, 2219). As shown in Fig. R12c, P-DY2DP-Por exhibits no detectable signal, in agreement with its insulating nature predicted by DFT. In stark contrast, DY2DP-Por shows a pronounced signal, several orders of magnitude higher, consistent with the presence of mobile charge carriers and the dispersive valence band structure.

These results suggest that the high conductivity of DY2DP-Por observed in this study likely arises from a synergistic effect of p-type doping, valence band dispersion, and interlayer electronic coupling. Given the uniqueness of this finding and the

remarkable photoconductivity response, we plan to further explore this phenomenon in a dedicated study, incorporating extended DFT analysis, OPTP measurements, and device integration (e.g., photodetector).

Fig. R12 | **a**, Calculated electronic band structure for P-DY2DP-Por (without pyridine). **b**, Calculated electronic band structure for DY2DP-Por (with pyridine). **c**, THz photoconductivity dynamics of P-DY2DP-Por and DY2DP-Por normalized by absorbed photon density.

Actions:

We have revised the manuscript for clarification at Line 335 on Page 15.

‘...The high σ of DY2DP-Por is likely attributed to pyridine doping during the synthesis...’

More details on the calculation of the channel geometry is needed. The Fig. 5 shows the crystal covered with as many as 8 contacts. What was the actual channel length of the crystal, BETWEEN individual contacts? The description of the fabrication is unclear. Ti and then Au were deposited onto the crystal and THEN platinum was

deposited between the crystal and the gold (how?)

Response: Following the reviewer's suggestion, we have added the detailed description of the channel length and the device fabrication in the revised manuscript. The channel length of DY2DP-Por crystal device was measured to be $\sim 0.33 \mu\text{m}$ ($\sim 0.11 \mu\text{m}$ between individual contacts) using the SEM image (Fig. 5a). Considering the rectangular geometry of the cross-section of DY2DP-Por crystal, its area was roughly estimated by calculating the product of width and height ($\sim 0.6 \times 0.5 \mu\text{m}^2$).

In our study, the DY2DP-Por crystal device was fabricated through electron beam lithography (EBL) and focused ion beam (FIB) at ultrahigh vacuum (Fig. R13). The crystals were first diluted and uniformly dispersed onto a clean silicon wafer with predefined alignment marks. A layer of PMMA was then spin-coated onto the wafer, serving as the electron beam resist. Crystal device fabrication was identified using an optical microscope. EBL at 30 kV was employed to pattern the selected crystals. After development, the exposed regions were subjected to magnetron sputtering to sequentially deposit a 5 nm titanium adhesion layer followed by 50 nm of gold. The unwanted resist and overlying metal were then removed using a standard lift-off process, resulting in the desired electrode pattern. As the height of the single crystals exceeded that of the deposited metal electrodes, direct electrical contact could not be achieved. To establish a reliable connection, Pt was deposited at the contact interface between the crystals and the electrodes using FIB deposition, thereby securing the crystals and completing the electrical circuit.

Fig. 5| a, SEM images of DY2DP-Por crystal device with Pt contacts.

Fig. R13 | Schematic diagram of DY2DP-Por crystal device fabrication procedure.

Actions:

a. We have added the above discussion in the revised manuscript at Line 328-330 on Page 14.

‘...L values were measured to be $\sim 0.33 \mu\text{m}$ for DY2DP-Por and $\sim 12.8 \mu\text{m}$ for GDY. Their A values were roughly estimated to be $\sim 0.3 \mu\text{m}^2$ for DY2DP-Por and $\sim 40.7 \mu\text{m}^2$ for GDY. ...’

b. Fig. 5 has been modified in the revised manuscript (Line 350, Page 15).

Fig. 5 | *Electronic property of DY2DP-Por crystal. a, SEM image of DY2DP-Por crystal device with Pt contacts. b, Temperature-dependent conductivities of DY2DP-Por crystal ranging from 300 to 50 K. Inset: The plot of $\ln(\sigma)$ versus the reciprocal of the temperature ($1/T$). c, Zabrodskii plots, $\ln(W)$ versus $\ln(T)$, where*

$$W = d(\ln\sigma)/d(\ln T).$$

- c. We have added the crystal device fabrication procedure in the revised Supplementary Information on Page 7.

‘Crystal device fabrication. The DY2DP-Por crystal device was fabricated through electron beam lithography (EBL) and focused ion beam (FIB) at ultrahigh vacuum (Supplementary Fig. 50). The crystals were first diluted and uniformly dispersed onto a clean silicon wafer with predefined alignment marks. A layer of PMMA was then spin-coated onto the wafer, serving as the electron beam resist. Crystal device fabrication was identified using an optical microscope. EBL at 30 kV was employed to pattern the selected crystals. After development, the exposed regions were subjected to magnetron sputtering to sequentially deposit a 5 nm titanium adhesion layer followed by 50 nm of gold. The unwanted resist and overlying metal were then removed using a standard lift-off process, resulting in the desired electrode pattern. As the height of the single crystals exceeded that of the deposited metal electrodes, direct electrical contact could not be achieved. To establish a reliable connection, platinum (Pt) was deposited at the contact interface between the crystals and the electrodes using FIB deposition, thereby securing the crystals and completing the electrical circuit.’

- d. Fig. R13 has been added as Supplementary Fig. 50 in the revised Supplementary Information (Page 59).

Comment 10. *The authors attribute the conductivity to interlayer electronic coupling. Is this based on an assumption that the c-axis is the fastest growth direction or the actual mapping of the crystallographic direction was done? This should at least be explained, and, ideally, the direction mapped crystallographically.*

Response: Thanks for the constructive comment. The crystallographic direction of DY2DP-Por crystals was determined by TEM and SAED analysis (Fig. R14). The SAED pattern reveals diffraction spots corresponding to the (001) plane along the long axis of the DY2DP-Por crystal, indicating that the crystal length aligns with the interlayer stacking direction (Fig. R14a). Furthermore, the high-resolution TEM image

shows lattice fringes corresponding to the (200) plane along the width of the crystal, confirming that the crystal width corresponds to the intralayer direction (Fig. R14b). The corresponding FFT pattern is consistent with the SAED data, further validating the crystal orientation. Note that the electronic measurements were performed along the long axis of the crystal. Thus, the measured conductivity reflects the charge transport along the interlayer direction.

Fig. R14 | **a**, TEM image of rod-shaped DY2DP-Por crystals. Inset: SAED pattern of DY2DP-Por crystals. **b**, TEM, high-resolution TEM and the FFT images (inset) of rod-shaped DY2DP-Por crystals.

Actions:

- a. We have added the above discussion in the revised Supplementary Information on Page 60.

‘The crystallographic direction of DY2DP-Por crystals was determined by TEM and SAED analysis (Supplementary Fig. 51). The SAED pattern reveals diffraction spots corresponding to the (001) plane along the long axis of the DY2DP-Por crystal, indicating that the crystal length aligns with the interlayer stacking direction (Supplementary Fig. 51a). Furthermore, the high-resolution TEM image shows lattice fringes corresponding to the (200) plane along the width of the crystal, confirming that the crystal width corresponds to the intralayer direction

(Supplementary Fig. 51b). The corresponding FFT pattern is consistent with the SAED data, further validating the crystal orientation. Note that the electronic measurements were performed along the long axis of the crystal. Thus, the measured conductivity reflects the charge transport along the interlayer direction.’

- b. Fig. R14 has been added as Supplementary Fig. 51 in the revised Supplementary Information (Page 60).

Other less critical points:

Comment 11. *The role of surface tension is not entirely clear. There are a number of other parameters which could explain why DMAc:H₂O works while other solvents don't (ability to form PFAS monolayers; hydrophobic effect, etc).*

Response: We appreciate the reviewer for the valuable comments. In our study, the formation of stable and crystalline PFS monolayer is critical for driving the accumulation of Cu⁺ and oriented crystal growth on the DMAc-H₂O surface. In light of this, we highlight that sufficient surface tension is a prerequisite for supporting amphiphilic PFS surfactants on the DMAc-H₂O surface, facilitating their self-assembly into stable monolayer that can guide the 2D polymerization (*Chem. Rev.* **2022**, 122, 6459-6513). In contrast, solvents with low surface tension (e.g., DMAc, CHCl₃, EtOH, 1,4-dioxane; typically < 35 mN m⁻¹) cannot stabilize PFS monolayer, leading to its diffusion into the bulk or precipitation (Fig. R15).

We acknowledge that surface tension alone cannot fully account for the selection of the DMAc-H₂O mixture. Other parameters, such as hydrogen-bonding capability, solvent polarity, and the hydrophobic effect, are likely to contribute too. For instance, DMAc-H₂O mixture has greater hydrogen-bonding capability at the surface than pure DMAc, due to the presence of water as a hydrogen-bond donor and acceptor, which help favor monolayer formation and structural ordering (*Angew. Chem. Int. Ed.* **1992**, 31, 130-152). However, direct experimental study of these individual contributions is currently challenging. Considering these complexities, we have revised the manuscript to clarify that a combination of interface properties (e.g., surface tension, interfacial hydrogen bonding, etc.) collectively governs the selection of DMAc-H₂O mixture as

solvent for O-SMAIS.

Fig. R15 Surface pressures of the DMAc, CHCl₃, EtOH and 1,4-dioxane surface after adding PFS molecules.

Actions:

- a. We have revised the manuscript at Line 113-115 on Page 4.

‘...A combination of interface properties (e.g., surface tension, interfacial hydrogen bonding, etc.) collectively governs the selection of a mixture of DMAc and H₂O (v:v=1:1, named as DMAc-H₂O) as solvent...’

- b. We have added the above discussion in the revised Supplementary Information on Page 13.

‘In our study, the formation of stable and crystalline PFS monolayer is critical for driving the accumulation of Cu⁺ and oriented crystal growth on the DMAc-H₂O surface. In light of this, we highlight that sufficient surface tension is a prerequisite

for supporting amphiphilic PFS surfactants on the DMAc-H₂O surface, facilitating their self-assembly into stable monolayer that can guide the 2D polymerization. In contrast, solvents with low surface tension (e.g., DMAc, CHCl₃, EtOH, 1,4-dioxane; typically < 35 mN m⁻¹) cannot stabilize PFS monolayer, leading to its diffusion into the bulk or precipitation (Supplementary Fig. 6).’

c. Fig. R15 has been added as Supplementary Fig. 6 in the revised Supplementary Information (Page 13)

Comment 12. The GISAXS data (Fig. 3d) is hard to read as it is dominated by the shadow of the beam knife. Consider adding arrows to point to the features of interest.

Response: We thank the reviewer for bringing this point to our attention. In the revised manuscript, the features in the GISAXS data have been marked with black arrows for clarity.

Fig. 3| d, Representative in-situ GISAXS patterns for Step I-III measured on the DMAc-H₂O surface.

Actions:

The features in Fig. 3d have been marked in the revised manuscript at Line 196 on Page 9.

Comment 13. The term "oxidative addition" may not be appropriate for the proposed mechanism given the CH bond is broken before the coordination with copper.

Response: We thank the reviewer for pointing out this mistake. Indeed, this oxidation refers to the oxidative coupling from the Cu⁺-acetylide complex to the dicopper²⁺-

diacetylide complex. We have corrected the discussion on the Glaser coupling mechanism in the revised manuscript.

Actions:

The discussion on Glaser coupling mechanism have been corrected in the revised manuscript at Line 167-168 on Page 7.

‘The Glaser coupling involves three steps: coordination (3 to 3a), oxidation (3a to 3b), and reductive elimination (3b to 4).’

Comment 14. The “horizontal transfer” of the polymer needs a clarification. Is this a Langmuir-Schaeffer type transfer (substrate touching the liquid surface)?

Response: We thank the reviewer for pointing this out. In our study, unlike Langmuir-Schaeffer type transfer, which involves horizontal contact between the substrate and the top surface of floating film, we employ a dipping process where the substrate (e.g., SiO₂/Si wafer, TEM grid) is carefully positioned beneath the 2DPs floating on the DMAc-H₂O surface and then lifted horizontally to pick them up (Fig. R16). To avoid ambiguity, we have clarified the term “horizontal transfer” in the Methods section in the revised Supplementary Information.

Fig. R16| Schematic illustration of the horizontal transfer process.

Actions:

a. We have added the above discussion in the revised manuscript at Line 138-140 on Page 5.

‘After 24 hours, we employ a dipping process where the substrate (e.g., SiO₂/Si

wafer, TEM grid) is carefully positioned beneath the 2DPs floating on the DMAc-H₂O surface and then lifted horizontally (Supplementary Fig. 7).’

- b. Fig. R16 has been added as Supplementary Fig. 7 in the revised Supplementary Information (Page 14).

Comment 15. Abbreviation *a.u.* is used for both arbitrary units (Figs. 2, 3) and atomic units (hartree, Fig. S16).

Response: We thank the reviewer for identifying this error. The unit of Y axis in Supplementary Fig. 23c has been corrected in the revised Supplementary Information.

Supplementary Fig. 23| c, The energy of TEPP molecules underneath a PFS monolayer with different angles.

Actions:

We have corrected the unit of Y axis in Supplementary Fig. 23c in the revised Supplementary Information on Page 31.

Comment 16. The Fig. S18 supporting the epitaxial relation between the PFS and the polymer is hard to decipher. Please add a side view. Also, this point may need additional discussion in the text. Given the low directionality of the electrostatic interaction, how is the epitaxy achieved, without significantly compromising the intra/inter-sheet interactions in the polymer?

Response: Thanks for the constructive comment. Indeed, Supplementary Fig. 25 is a simple lattice superposition between the obtained intralayer structure of a PFS monolayer and the interlayer structure of DY2DP-Por crystals. To improve clarity, we

have now included both top and side views of the PFS monolayer and the DY2DP-Por crystal in Fig. R17. A comparative analysis of these two lattice structures reveals partial crystallographic alignment, such as the (100) plane of the PFS monolayer (d -spacing of ~ 0.50 nm) show lattice matching with the (040) plane of DY2DP-Por (d -spacing of ~ 0.50 nm). As previously reported (*Nat. Commun.* **2023**, 14, 8313), this observation supports the notion that DY2DP-Por crystals grow epitaxially beneath the long-range ordered PFS monolayer.

We agree that electrostatic interactions generally exhibit low directionality and are insufficient to drive epitaxial alignment. In our system, the role of electrostatic interaction is to guide the accumulation of Cu^+ ions underneath the PFS monolayer, forming the Cu^+ -rich DMAc- H_2O surface. The spatially confined Cu^+ ions then act as coordination centers that direct the monomer adsorption and assembly via the formation of coordination bonds between Cu^+ ions and terminal acetylene groups, which are known for their directional nature. This coordination-driven assembly can promote crystallographic alignment during polymerization, thereby facilitating epitaxial growth without disrupting the intrinsic intra- and interlayer interactions within the polymer lattice. In the revised manuscript, we have added more details to elaborate this coordination-driven epitaxial growth process.

Fig. R17 | Lattice structures of the PFS monolayer and the DY2DP-Por crystal, and a simple lattice superposition of their top views.

Actions:

- a. We have added more details in the revised manuscript at Line 240-245 on Page 11.
'...A comparative analysis of these two lattice structures reveals partial crystallographic alignment, such as the (100) plane of the PFS monolayer (d-spacing of ~0.50 nm) show lattice matching with the (040) plane of DY2DP-Por (d-spacing of ~0.50 nm) (Supplementary Fig. 25). This observation supports the notion that DY2DP-Por crystals grow epitaxially beneath the long-range ordered PFS monolayer (Supplementary Figs. 26-30).'
- b. We have added the above discussion in the revised Supplementary Information on Page 34.
'In our system, the role of electrostatic interaction is to guide the accumulation of Cu⁺ ions underneath the PFS monolayer, forming the Cu⁺-rich DMAc-H₂O surface. The spatially confined Cu⁺ ions then act as coordination centers that direct the monomer adsorption and assembly via the formation of coordination bonds between Cu⁺ ions and terminal acetylene groups, which are known for their directional nature. This coordination-driven assembly can promote crystallographic alignment during polymerization, thereby facilitating epitaxial growth without disrupting the intrinsic intra- and interlayer interactions within the polymer lattice.'
- c. Fig. R17 has been added as Supplementary Fig. 25 in the revised Supplementary Information (Page 34)

Comment 17. *The XPS Fig. S25 show a very significant amount of C=O (double the amount of CN). Where is this coming from? I do not understand the deconvolution of N1s. What's the nature of the third component and what's the meaning of pyrrolic N-Cu vs pyridinic N-Cu (if Cu is incorporated in the porphyrine, all 4 nitrogens become identical). Also a survey spectrum showing all observed elements (Cu, F, Si, etc.) is needed.*

Response: We thank the reviewer for the valuable comment. We attribute the observed

C=O signal in the XPS spectra to surface-adsorbed carbon dioxide and oxygen molecules, which may interact with the samples surface through either physisorption or chemisorption. Such adsorption is well-documented under ambient conditions and is known to contribute to C=O signatures in XPS spectra (*Chem. Commun.* **2015**, 51, 1834-1837; *J. Am. Chem. Soc.* **2017**, 139, 3145-3152). Representative XPS C1s spectra of graphdiyne from the literature showing similar CO₂-related features have been included in Fig. R18 for comparison. Considering that the detection depth of XPS is typically limited to only a few nanometers, even an ultrathin layer of adsorbed species or surface oxides can produce a pronounced signal in the measurement.

Regarding the N 1s spectrum, we agree that all the 4 N in the porphyrin macrocycle feature identical peaks. The peaks observed at 401.1 eV, 399.3 eV, and 398.0 eV correspond to N coordinated to Cu in the porphyrin macrocycle ((pyrrolic N)₄-Cu), imine nitrogen (C=N) in both porphyrin and pyridine molecules, and pyridinic N coordinated to Cu (pyridinic N-Cu), respectively. To avoid ambiguity, the corresponding chemical structures have been included in Supplementary Fig. 25. Following the reviewer's suggestion, the XPS survey spectrum of DY2DP-Por is now provided in Fig. R19, which confirms the presence of all expected elements (C, N, Cu).

Fig. R18 | **a**, XPS survey scan of graphdiyne. **b**, Curve-fitted high-resolution XPS C 1s spectra of graphdiyne (*Chem. Commun.* **2015**, 51, 1834-1837).

Supplementary Fig. 25| b, Curve-fitted high-resolution XPS N 1s spectra of DY2DP-Por crystals.

Fig. R19| XPS survey scan of DY2DP-Por crystals.

Actions:

- a. We have added the above discussion in the revised Supplementary Information on Page 43.

‘We attribute the observed C=O signal in the XPS spectra to surface-adsorbed carbon dioxide and oxygen molecules, which may interact with the samples surface through either physisorption or chemisorption. Considering that the detection depth of XPS is typically limited to only a few nanometers, even an ultrathin layer of adsorbed species or surface oxides can produce a pronounced signal in the

measurement.

Regarding the N 1s spectrum, all the 4 N in the porphyrin macrocycle feature identical peaks. The peaks observed at 401.1 eV, 399.3 eV, and 398.0 eV correspond to N coordinated to Cu in the porphyrin macrocycle ((pyrrolic N)₄-Cu), imine nitrogen (C=N) in both porphyrin and pyridine molecules, and pyridinic N coordinated to Cu (pyridinic N-Cu), respectively. To avoid ambiguity, the corresponding chemical structures have been included in Supplementary Fig. 35b. The XPS survey spectrum of DY2DP-Por is now provided in Supplementary Fig. 34, which confirms the presence of all expected elements (C, N, Cu)'

- b. We have modified the Supplementary Fig. 35 in the revised Supplementary Information on Page 44.
- c. Fig. R19 has been added as Supplementary Fig. 34 in the revised Supplementary Information (Page 43).

Comment 18. *The volume of Cu and monomer solutions injected in the petri dish should be specified.*

Response: We thank the reviewer for bringing this mistake to our attention. We have added the volume (i.e., 1 ml) of CuCl and monomer solutions in the Methods section in the revised manuscript.

Actions:

The volumes of CuCl and monomer solutions have been added in the Methods section in the revised manuscript at Line 380, Page 16 and Line 389, Page 17.

In short, we would like to thank all the reviewers for the constructive and valuable comments and the time to handle our manuscript, which have helped us to substantially improve the quality of the work.

Point-to-point response to the comments from the reviewers

Reviewer #1

General comment. *The authors addressed satisfactorily the issues I raised in the previous round of reviews. Following the changes I can recommend publication of the current manuscript.*

Response: We appreciate Reviewer#1 for the encouraging comments and the positive recommendation for publication.

Reviewer #2

General comment. *The authors have carefully addressed all the comments and revised the manuscript thoroughly.*

Response: We appreciate Reviewer#2 for the positive recommendation for publication.

Reviewer #3

General comment. *The authors provided a remarkably thorough revision and replied to all of my comments in full. I am pleased to recommend the paper for acceptance.*

Response: We sincerely thank Reviewer#3 for the valuable comments and positive recommendation.

Comment 1. *At the author's discretion (no further review is need), I suggest to revise the discussion of N1s N deconvolution in the SI (P.43, Fig. S25). The labelling of "imine" nitrogen seems confusing as there's no imine group in the structure. Possibly, the authors refer to the pyridinic nitrogen of the uncoordinated porphyrine ring, but that would imply that most of the porphyrine rings are uncoordinated. Also, the ratio of peaks assigned to Cu-porphyrine vs Cu-Py is far from the expected 4:1. In my opinion, the more likely scenario is that the 399 eV peak is from Cu-porphyrine, 401 eV is from Cu-Py (expected at higher BE based on electrostatics) and the 398 eV peaks is the pyrrolic N in the uncoordinated H₂-porphyrine. The author could compare the elemental ratio of Cu vs (deconvoluted) N to better support the assignment.*

Response: Following the reviewer's suggestion, we have revised the figure and discussion of the XPS N1s spectra in the revised Supplementary Information (Supplementary Fig. 35b). The peaks observed at 401.1 eV, 399.3 eV, and 398.0 eV correspond to the pyridinic N coordinated to Cu (pyridinic N-Cu), N coordinated to Cu in the porphyrin macrocycle ((pyrrolic N)₄-Cu), and the pyrrolic N in the uncoordinated H₂-porphyrin macrocycle, respectively. Quantitative analysis reveals that the ratios of (pyrrolic N)₄-Cu and pyridinic N-Cu peaks are 7:3, suggesting

that the Cu is coordinated at the center of the porphyrin macrocycle with two pyridine molecules serving as axial ligands positioned above and below the porphyrin plane.

Supplementary Fig. 35| b, Curve-fitted high-resolution XPS N 1s spectra of DY2DP-Por crystals.

Actions:

We have added the above discussion in the revised Supplementary Information on Page 43.

'...The peaks observed at 401.1 eV, 399.3 eV, and 398.0 eV correspond to the pyridinic N coordinated to Cu (pyridinic N-Cu), N coordinated to Cu in the porphyrin macrocycle ((pyrrolic N)₄-Cu), and the pyrrolic N in the uncoordinated H₂-porphyrin macrocycle, respectively. Quantitative analysis reveals that the ratios of (pyrrolic N)₄-Cu and pyridinic N-Cu peaks are 7:3, suggesting that the Cu is coordinated at the center of the porphyrin macrocycle with two pyridine molecules serving as axial ligands positioned above and below the porphyrin plane....'